# SVL: Goal-Conditioned Reinforcement Learning as Survival Learning

**Franki Nguimatsia Tiofack** [1]   **Fabian Schramm** [1]   **Théotime Le Hellard** [1]   **Justin Carpentier** [1]

## Abstract

Standard approaches to goal-conditioned reinforcement learning (GCRL) that rely on temporal-difference learning can be unstable and sample-inefficient due to bootstrapping. While recent work has explored contrastive and supervised formulations to improve stability, we present a probabilistic alternative, called survival value learning (SVL), that reframes GCRL as a survival learning problem by modeling the distribution of time-to-goal from each state. This structured distributional Monte Carlo perspective yields a closed-form identity that expresses the goal-conditioned value function as a discounted sum of survival probabilities, enabling value estimation via a hazard model trained via maximum likelihood on both event and right-censored trajectories. We introduce three practical value estimators, including finite-horizon truncation and two binned infinite-horizon approximations to capture long-horizon objectives. Experiments on offline GCRL benchmarks show that SVL combined with hierarchical actors matches or surpasses strong hierarchical TD and Monte Carlo baselines, excelling on complex, long-horizon tasks. Webpage and code

## 1. Introduction

Goal-Conditioned Reinforcement Learning (GCRL) has emerged as an interesting paradigm for building generalist agents that learn policies capable of reliably reaching a wide range of goals from diverse initial states. Rather than relying on carefully shaped reward functions, GCRL formulates tasks directly in terms of desired outcomes, such as target positions or object configurations. This is appealing for autonomous agents and robotic systems, where specifying goals is often more natural than designing dense reward

[1]Inria and École Normale Supérieure, PSL Research University, Paris, France. Correspondence to: Franki Nguimatsia Tiofack <franki.nguimatsia-tiofack@inria.fr>.

*Proceedings of the 43rd International Conference on Machine Learning*, Seoul, South Korea. PMLR 306, 2026. Copyright 2026 by the author(s).

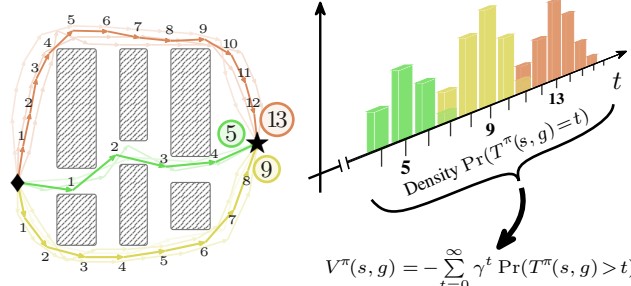

*Figure 1.* **Illustration of time-to-goal distribution models in GCRL.** In the general GCRL setting, the time required to reach a goal is a random variable. As illustrated in the navigation task (left), an agent may reach the target (★) from the start (♦) via multiple distinct paths, resulting in a multi-modal distribution of arrival times (right) with modes at $t \approx 5, 9, 13$. Traditional TD learning approaches estimate the goal-conditioned value function by backpropagating reward signals through all transitions from the goal to the state. Instead, we propose directly exploiting the arrival-time realizations (colored circles). Our survival value learning (SVL) approach models this distribution and recovers the value function $V^\pi(s, g)$ as a discounted sum of survival probabilities.

signals. Hence, most GCRL works rely on binary signals indicating success only upon achieving the goal.

While conceptually appealing, sparse rewards pose notable optimization challenges. Standard approaches typically mitigate sparsity via hindsight relabeling (Andrychowicz et al., 2017; Ding et al., 2019), which convert each trajectory into multiple goal-labeled transitions by treating achieved states as alternative goals. However, methods that rely on Temporal Difference (TD) learning inherit the fundamental instability of bootstrapping. In long-horizon settings common in GCRL, error propagation can make TD learning slow, unstable, and hyperparameter-sensitive. Recent Monte Carlo (MC) works sought to avoid bootstrapping by exploiting the structure of goal-reaching problems. Prior approaches reformulate GCRL objectives using supervised learning (Ghosh et al., 2021; Yang et al., 2022; Hejna et al., 2023) or contrastive learning (Eysenbach et al., 2022), enabling more stable optimization.

In this work, we introduce a complementary probabilistic perspective that interprets GCRL through the lens of time-to-event modeling, also known as survival analysis (Lee & Wang, 2003; Wang et al., 2019). This statistical framework studies the distributions of variables representing

the time until a specific event occurs. Our key observation is that in sparse, terminate-on-success settings, the expected discounted return is fully determined by the distribution of the goal-hitting time under the policy. Maximizing the return thus corresponds to optimizing a discounted functional of the hitting-time distribution. The discount factor $\gamma$ induces an implicit risk sensitivity over arrival times. Unlike in standard RL, intermediate states in GCRL settings do not yield additional rewards; the value function is therefore a statistical summary of the arrival-time distribution and can be estimated directly, as illustrated in Fig. 1. Successful trajectories yield exact event times, whereas those truncated by a finite horizon become right-censored observations. This correspondence naturally aligns policy evaluation in GCRL with the methodological foundations of survival analysis.

In this work, we derive a closed-form expression of the goal-conditioned value function as a discounted sum of survival probabilities associated with the goal-hitting time. We show that the value function is a statistical summary of an underlying temporal distribution. Consequently, rather than directly regressing a scalar value via Bellman backups, we explicitly learn the full distribution of goal-reaching times, naming our approach **Survival Value Learning (SVL)**. We model the instantaneous probability of reaching the goal, namely the hazard function, via maximum likelihood estimation (MLE) over both event and censored trajectories, thereby inheriting standard MLE convergence properties.

SVL incorporates hindsight relabeling to convert each trajectory into multiple goal-labeled survival observations, providing dense supervision without altering the underlying objective. Unlike prior work using relabeling to train value functions (Andrychowicz et al., 2017; Ren et al., 2019; Zhao & Tresp, 2018; Fang et al., 2019; Park et al., 2023; Yang et al., 2022; Hejna et al., 2023) or representations (Eysenbach et al., 2022; Mazoure et al., 2023; Sikchi et al., 2023), we use it to generate event-time and censoring signals.

Our key contributions are: (i) we introduce SVL, a theoretical framework connecting GCRL with survival analysis. We propose a practical architecture to learn the hazard function and derive several value estimators trained via maximum likelihood, bypassing unstable TD bootstrapping. (ii) By integrating our SVL estimators with hierarchical actors, we introduce Hierarchical Survival Value Learning (HSVL), a practical offline GCRL algorithm. (iii) Through extensive evaluations and ablation studies on offline GCRL benchmarks, we show that survival-based value estimation is competitive with strong hierarchical TD and MC baselines and delivers notable improvements on long-horizon tasks, without additional hyperparameter tuning.

The remainder is organized as follows. Sec. 2 reviews related work, and Sec. 3 introduces background on goal-conditioned reinforcement learning and survival analysis notations. Sec. 4 presents the core contribution, establishing the connection between GCRL and survival learning and introducing the corresponding value estimators inspired by survival analysis. Sec. 5 evaluates the proposed estimators on standard offline RL benchmarks, and Sec. 6 draws conclusions and future work.

## 2. Related work

### 2.1. Goal-conditioned reinforcement learning

The problem of learning policies for multiple goals was formalized by universal value function approximators (Schaul et al., 2015), which extended value functions to accept goal arguments. To mitigate the reward sparsity of goal-reaching tasks, Hindsight Experience Replay (Andrychowicz et al., 2017) proposed relabeling failed trajectories as successful ones for the states that were actually reached. Since then, numerous works have studied GCRL primarily through temporal-difference learning and goal-conditioned value estimation (Andrychowicz et al., 2017; Lin et al., 2019; Riedmiller et al., 2018; Park et al., 2023; Ke et al., 2025; Ahn et al., 2025).

Beyond TD-based approaches, alternative formulations have been explored. Imitation and behavioral cloning methods leverage goal-conditioned supervision from trajectories (Ding et al., 2019; Ghosh et al., 2021; Lynch et al., 2020; Savinov et al., 2018; Jang et al., 2022). Model-based approaches incorporate planning or learned dynamics to guide goal-reaching behaviour (Mendonca et al., 2021; Charlesworth & Montana, 2020; Zhu et al., 2021; Dosovitskiy & Koltun, 2017; Schmeckpeper et al., 2020). A complementary line of work combines learned value functions with graph search over replay-buffer (Eysenbach et al., 2019; Baek et al., 2025). Representation learning methods shape exploration and goal achievement by learning goal-conditioned embeddings, successor-style representations, or by modeling goal-conditioned value functions through contrastive learning (Barreto et al., 2017; Hansen et al., 2020; Eysenbach et al., 2021; 2022; Liu & Abbeel, 2021; Blier et al., 2021; Wang et al., 2025) and Ma et al. (2022) recast offline GCRL as matching a discounted state-visitation distribution. Some works combine multiple approaches; for example, offline GCRL methods often integrate TD learning with hindsight relabeling to address data scarcity (Park et al., 2023; Ke et al., 2025), while contrastive RL approaches couple representation learning with relabeling to learn goal-conditioned representations without relying on a TD objective (Eysenbach et al., 2022). Supervised and imitation-based formulations similarly use hindsight relabeling to construct dense training targets from sparse trajectories (Ghosh et al., 2021; Emmons et al., 2022).

Closest to our work is distance-weighted supervised learning

(DWSL) (Hejna et al., 2023), which also models the distribution of number of step between state pairs in a Monte-Carlo, non-TD fashion. SVL differs as it handles unreached goals via right censoring, rather than restricting supervision to state pairs from the same trajectory and recovers the goal-conditioned value function in closed form from the learned survival function, instead of relying on a soft-minimum distance surrogate that lower-bounds the value function.

## 2.2. Survival analysis and time-to-event learning

Survival analysis (Lee & Wang, 2003), also known as time-to-event analysis, is a classical area of statistics concerned with modeling the distribution of event times, typically through survival and hazard functions (Kaplan & Meier, 1958; Cox, 1972; Nelson, 1969; Andersen et al., 2012; Tutz et al., 2016). It has been widely applied in medicine and epidemiology (e.g, time to death), reliability engineering (time to failure), and the social sciences (event-history analysis) (Kalbfleisch & Prentice, 2002; Singer & Willett, 2003). In this statistical branch, foundational tools include nonparametric estimators (Kaplan & Meier, 1958; Andersen et al., 2012), semi-parametric regression models such as the Cox proportional hazards model (Cox, 1972), discrete-time or grouped-time formulations (Tutz et al., 2016; Singer & Willett, 2003), which we leverage in this work. These models come with a well-developed theory for censored likelihoods and asymptotic guarantees (Kalbfleisch & Prentice, 2002; Andersen et al., 2012; Fleming & Harrington, 2013).

More recently, survival analysis has been extended using deep learning, including neural Cox models and direct hazard or survival function learning (Katzman et al., 2018; Kvamme et al., 2019; Ranganath et al., 2016; Lee et al., 2018). Our work builds on this literature by interpreting goal-reaching rollouts as survival data. Unlike prior alternative discounting schemes (Schultheis et al., 2022), formulated via policy-independent survival terminology (Fedus et al., 2019), SVL uses the standard discounted GCRL objective to model the policy-dependent first-hitting-time distribution.

# 3. Preliminaries

This section provides an overview of GCRL and survival analysis, and introduces the main notations of the paper.

## 3.1. Goal-conditioned reinforcement learning

The GCRL problem is formulated as an augmented Markov Decision process (MDP), defined by the tuple $\mathcal{M} = (\mathcal{S}, \mathcal{A}, \mathcal{G}, \mathcal{P}, r, p_g, \mu, \phi, \gamma)$. Here, $\mathcal{S}$, $\mathcal{A}$, and $\mathcal{G}$ denote the state, action, and goal spaces, respectively. Let $\Delta(\mathcal{X})$ represent the set of probability distributions over a set $\mathcal{X}$. $\mathcal{P} : \mathcal{S} \times \mathcal{A} \to \Delta(\mathcal{S})$ represents the transition dynamic

and $r : \mathcal{S} \times \mathcal{A} \times \mathcal{G} \to \mathbb{R}$ is the goal-conditioned reward function. Desired goals are sampled from the distribution $p_g \in \Delta(\mathcal{G})$ with initial states drawn from $\mu : \mathcal{G} \to \Delta(\mathcal{S})$. Finally, $\gamma \in [0, 1)$ is the discount factor and $\phi : \mathcal{S} \to \mathcal{G}$ is a tractable mapping that projects the state into goal space. This mapping accounts for settings where goals represent only partial observations or specific features of the state. For example, in robotic manipulation, $\mathcal{S}$ might represent joint angles while $\mathcal{G}$ represents end-effector coordinates. In this case, $\phi$ corresponds to the forward kinematics function.

Following prior work (Andrychowicz et al., 2017; Liu et al., 2022), we adopt a sparse reward formulation in which $r(s_t, a_t, g) = -\mathbf{1}_{\{\|\phi(s_{t+1})-g\| \geq \epsilon\}}$. That is, the agent receives a penalty of $-1$ whenever the next state $s_{t+1}$ fails to satisfy the goal $g$ (within tolerance $\epsilon$). The objective of GCRL is to learn a goal-conditioned policy, denoted by $\pi : \mathcal{S} \times \mathcal{G} \to \Delta(\mathcal{A})$, that maximizes the expected discounted cumulative return:

$$J(\pi) = \mathbb{E}_{g \sim p_g, \ \tau \sim \rho^\pi(\cdot|g)}\left[\sum_{t=0}^{\infty} \gamma^t r(s_t, a_t, g)\right], \quad (1)$$

where $\rho^\pi(\tau|g) = \mu(s_0|g) \prod_{t=0}^{\infty} \pi(a_t|s_t, g) \mathcal{P}(s_{t+1}|s_t, a_t)$ is the probability of sampling a trajectory $\tau = (s_0, a_0, s_1, a_1, \dots)$. Intuitively, this objective corresponds to sampling a goal $g$ and then optimizing the policy to reach that goal as quickly as possible (Eysenbach et al., 2022). Accordingly, for a given policy $\pi$, we denote the goal-conditioned value-function as follows:

$$V^\pi(s, g) = \mathbb{E}_{\tau \sim \rho^\pi(\cdot|g)}\left[\sum_{t=0}^{\infty} \gamma^t r(s_t, a_t, g)|_{s_0=s}\right]. \quad (2)$$

## 3.2. Survival analysis notations in the context of GCRL

**First hitting time.** In analogy with survival analysis (Kaplan & Meier, 1958; Katzman et al., 2018; Kvamme et al., 2019), given a state $s$, a goal $g$, and a goal-conditioned policy $\pi$, we define the *first hitting time* to reach the goal as:

$$T^\pi(s, g) = \inf\left\{ t \geq 0 : \|\phi(s_{t+1})-g\| < \epsilon \ \middle| \ {}^{s_0=s, \ a_t \sim \pi(\cdot|s_t, g)}_{s_{t+1} \sim \mathcal{P}(\cdot|s_t, a_t)} \right\}. \quad (3)$$

We adopt the convention $T^\pi(s, g) = \infty$ if the goal $g$ is never reached from the initial state $s$. Thus, $T^\pi(s, g)$ is the (possibly infinite) time of first arrival to $g$ when starting from $s$ and following $\pi$ under dynamics $\mathcal{P}$. Unlike standard survival analysis, where event times typically start at 1, we allow $t = 0$ to account for instant hits, in which the goal is satisfied immediately after the initial state $s_0$.

Formally, for a fixed goal and policy $(g, \pi)$, the system evolves as a Markov chain on states $(s_t)_{t \geq 0}$, defined by the transition kernel:

$$\mathcal{P}^{\pi, g}(s'|s) := \int_{\mathcal{A}} \pi(a|s, g) \, \mathcal{P}(s'|s, a) \, da. \quad (4)$$

The goal hitting time $T^\pi(s, g)$ is a stopping time with respect to the natural filtration of the process $(s_t)_{t \geq 0}$.

**Survival and hazard functions.** As in standard discrete time-to-event modeling (Tutz et al., 2016) and survival analysis more broadly (Andersen et al., 2012), we study $T^\pi(s, g)$ through its associated survival and hazard functions, which provide an equivalent characterization of its distribution. The survival function is defined by

$$S^\pi(t|s, g) = \Pr\big(T^\pi(s, g) > t\big), \qquad (5)$$

and corresponds to the probability that the goal has not been reached by time $t$. The discrete-time hazard function is

$$h^\pi(t|s, g) = \Pr\Big(T^\pi(s, g) = t \;\Big|\; T^\pi(s, g) \geq t\Big). \quad (6)$$

It measures the likelihood of reaching the goal at time $t$ given that it has not yet been reached. Note that the hazards fully determine the law of $T^\pi(s, g)$ as

$$S^\pi(t|s, g) = \prod_{k=0}^{t} \big(1 - h^\pi(k|s, g)\big), \qquad (7)$$

this recursion is detailed in Appendix A.1. This yields the event-time probabilities

$$\Pr\big(T^\pi(s, g) = t\big) = h^\pi(t|s, g)S^\pi(t - 1|s, g). \quad (8)$$

**Episode termination.** In practice, rollouts are truncated after a finite horizon $c$ (e.g., due to timeout, or episode termination). If the goal is not reached by time $c$, we only observe that $T^\pi(s, g) > c$, providing no information about subsequent steps. This corresponds to *right censoring* at time $c$ (Nelson, 1972; Efron, 1977; Harrington & Fleming, 1982; Peto & Peto, 1972; Cox, 1972; Mantel et al., 1966). Mathematically, a censored observation contributes to the likelihood via the survival function $S^\pi(c|s, g)$, whereas an uncensored observation (goal reached at $t \leq c$) contributes via the event probability $\Pr(T^\pi(s, g) = t)$.

## 4. Goal-Conditioned RL as Survival Learning

This section draws the connection between GCRL and survival learning, and introduces our Survival Value Learning (SVL) approach. We first establish in Proposition 4.1 a link between the goal-conditioned value function and the survival function of the goal-hitting time. Then, by exploiting the survival-hazard recursion together with the event-time probabilities in Eq. (8), we derive a log-likelihood objective for learning a goal-conditioned hazard function. We address practical considerations and present tractable hazard-function parameterizations, along with plug-in estimators that recover goal-conditioned values from the learned hazards. Finally, we derive an offline GCRL algorithm to empirically evaluate the proposed framework.

### 4.1. Relating goal-conditioned value functions to survival functions

We formalize the link between GCRL and survival analysis. Under the standard sparse-reward formulation, the goal-conditioned value function admits a closed-form expression in terms of the survival function of the goal-hitting time.

**Proposition 4.1.** *Consider the GCRL setting where the agent receives a per-step penalty until success, $r(s_t, a_t, g) = -\mathbf{1}_{\{\|\phi(s_{t+1}) - g\| \geq \epsilon\}}$, and the episode terminates upon goal achievement. The goal-conditioned value function is equal to the negative discounted sum of the survival function:*

$$V^\pi(s, g) = -\sum_{t=0}^{\infty} \gamma^t S^\pi(t|s, g). \qquad (9)$$

*Proof.* We refer readers to Appendix A.2.
Prop. 4.1 is exact when the return is fully determined by the goal-hitting time $T^\pi(s, g)$. For arbitrary dense rewards that depend on the full trajectory, the return is not identifiable from the distribution of $T^\pi(s, g)$ alone.

This identity implies that estimating the survival function (equivalently, the hazard function) yields a direct approximation of the value function. It offers an intuitive interpretation of the objective: maximizing $V^\pi(s, g)$ amounts to minimizing the discounted cumulative probability of *not* having reached the goal over time. In other words, this incentivizes the agent to minimize the task's survival time, with the discount factor giving greater weight to earlier successes.

### 4.2. Learning the hazard function

We derive a population objective for learning hazards, deferring modeling and approximation choices to Section 4.3.

**Observation.** Lets consider a sample $(g, s) \sim p_g \times \mu$ and roll out $\pi(\cdot|s_t, g)$ under dynamics $\mathcal{P}$ for a horizon $c$. Let $\delta \in \{0, 1\}$ indicate whether the goal is reached within $c$ steps, and let $\tau$ denote the (first) hitting time when $\delta = 1$. Thus, $\delta = 1$ implies $\tau \leq c$ (uncensored), while $\delta = 0$ corresponds to right censoring at $c$. The likelihood of an observation $(\tau, c, \delta)$ is then given by

$$\Pr(\tau, c, \delta|s, g) = \Pr\big(T^\pi(s, g) = \tau\big)^\delta S^\pi(c|s, g)^{1-\delta} \quad (10)$$

**Parametric hazard and negative log-likelihood.** Let $h_\theta^\pi(t|s, g)$ be a parametric hazard model parameterized by $\theta$. Using the standard discrete-time identities provided in Eq. (7) and Eq. (8), the maximum likelihood estimation (MLE) amounts to minimizing the population negative log-likelihood $\mathcal{L}(\theta) = -\mathbb{E}[\log \Pr_\theta(\tau, c, \delta, s, g)]$, expanding to

$$\mathcal{L}(\theta) = -\mathbb{E}_{(s,g,\tau,c,\delta)}\Big[\delta\ell_\tau(\theta) + (1 - \delta)\ell_c(\theta)\Big] \qquad (11)$$

with

$$\ell_\tau(\theta) = \log h_\theta^\pi(\tau|s,g) + \sum_{k=0}^{\tau-1} \log\left(1 - h_\theta^\pi(k|s,g)\right), \quad (12)$$

$$\ell_c(\theta) = \sum_{k=0}^{c} \log\left(1 - h_\theta^\pi(k|s,g)\right). \quad (13)$$

It is worth noting that the expectation is taken with respect to the tuple $(\tau, c, \delta, s, g)$, using the identity $\Pr_\theta(\tau, c, \delta, s, g) = p_g(g)\,\mu(s|g)\,\Pr_\theta(\tau, c, \delta|s, g)$.

**Censoring and empirical risk minimization.** In GCRL, censoring is induced by the episode horizon $c$. Since the horizon is determined by the environment configuration (or sampling strategy) and is independent of $T^\pi(s, g)$, it satisfies the condition of non-informative right censoring; informative terminations such as catastrophic failures fall outside this assumption and instead call for a competing-risks formulation (Andersen et al., 2012), which we leave to future work. As in standard RL, we optimize an empirical objective by sampling tuples $(s_i, g_i, \tau_i, c_i, \delta_i)$ either from an offline dataset or a replay buffer, which we treat as approximately i.i.d.

**Lemma 4.2** (MLE/ERM properties). *Assume (i) non-informative right censoring, and (ii) standard regularity conditions for parametric MLE (correct specification, identifiability, and smoothness of $h_\theta$). Let $\theta^*$ be the true parameter (i.e, $h_{\theta*}^\pi = h^\pi$, with $h^\pi$ defined in Eq. (6)) and*

$$\hat{\theta}_n \in \arg\min_{\theta \in \Theta}\left\{-\frac{1}{n}\sum_{i=1}^{n} \log \Pr_\theta(\tau_i, c_i, \delta_i, s_i, g_i)\right\}. \quad (14)$$

*Then $\hat{\theta}_n$ is consistent for $\theta^\star$ (under correct specification, or the KL minimizer in case of misspecification), and is asymptotically normal: $\sqrt{n}(\hat{\theta}_n - \theta^\star) \Rightarrow \mathcal{N}(0, \Sigma)$. Under correct specification, $\Sigma$ equals the inverse Fisher information.*

Lemma 4.2 implies that, as the number of survival tuples grows, the estimated hazard (and thus the estimated value) concentrates around its true value, with statistical fluctuations that scale as $\mathcal{O}(n^{-1/2})$ under standard conditions. In Section 5, we highlight this sample scaling property on the long-horizon humanoidmaze-giant task.

For standard references regarding Lemma 4.2 in the presence of right censoring, we refer readers to (Andersen et al., 2012) (Chapter 6),(Tutz et al., 2016) (Chapter 3), together with MLE theory (Fisher, 1925; Wald, 1949; Cramér, 1999).

**Hindsight relabeling.** Directly minimizing the empirical risk in Eq. 14 using only desired goals $g \sim p_g$ can be data-inefficient, since most rollouts may not reach the specified goal and thus provide weak supervision. Following prior work on goal relabeling (Andrychowicz et al., 2017; Park et al., 2023; Mazoure et al., 2023), we incorporate hindsight goal relabeling by treating visited states as additional

achieved goals. This transforms each trajectory into many goal-labeled survival tuples $(s_i, g_i, \tau_i, c_i, \delta_i)$, increasing the amount of event-time and right-censoring supervision available for maximum-likelihood hazard learning.

**Value estimation.** Given $\hat{\theta}_n$, Proposition 4.1 yields the plug-in estimator $V_{\hat{\theta}_n}^\pi(s, g) = -\sum_{t\geq 0} \gamma^t S_{\hat{\theta}_n}^\pi(t|s, g)$.

Our SVL approach provides an alternative estimator of the goal-conditioned value function that does not rely on bootstrapping. Instead, it follows from maximum-likelihood estimation of the hazard model and inherits the classical guarantees of MLE under standard regularity conditions (e.g., consistency and asymptotic normality).

### 4.3. Practical value estimation and hazard function architecture

In this section, we address two practical challenges to approximate the plug-in estimator $V_{\hat{\theta}_n}^\pi$ defined in Proposition 4.1: how to model the hazard function and how to handle the infinite discounted sum.

**Finite-horizon MDP.** In the finite-horizon setting of length $H$, the identity in Eq. (9) is a finite sum, yielding:

$$V_{\hat{\theta}_n}^{\pi, H}(s, g) = -\sum_{t=0}^{H-1} \gamma^t S_{\hat{\theta}_n}^\pi(t|s, g) \quad (15)$$

The straightforward approach is to model the hazard function with a neural network predicting an $H$-dimensional vector $\{h_\theta^\pi(t|s, g)\}_{t=0}^{H-1}$. Note that $\hat{\theta}_n$ is an optimal estimator, while $\theta$ are learned parameters. Each entry is the conditional probability of reaching the goal at time $t$, with $t = 0 \ldots H - 1$, given that it was not reached earlier. In that case, the training objective Eq. (11) remains unchanged.

Yet, this approach does not extend directly to the infinite-horizon settings, as it ignores any value accumulated beyond the horizon $H$. Furthermore, it requires fixing $H$ a priori, and optimization becomes increasingly difficult as the horizon lengthens. To address these limitations while targeting infinite-horizon settings, we propose reducing the output dimensionality by aggregating time steps into intervals.

**Geometric-time binning.** To reduce complexity while preserving resolution at early time steps, we adopt grouped-time modeling from discrete-time survival analysis (Tutz et al., 2016). Let $K$ be the number of bins, $0 = b_0 < b_1 < \cdots < b_K = H$ be bin edges with intervals $[b_k, b_{k+1})$ and lengths $L_k = b_{k+1} - b_k$. We use geometric edges $b_k = \lfloor \rho^k \rfloor$, $\rho = H^{1/K}$ which allocates many short bins early (where $\gamma^t$ is large) and fewer long bins late (where the discounted tail contributes less). The hazard network is now designed to output $K + 1 < H$ logits.

Based on geometric time binning, we consider two binned

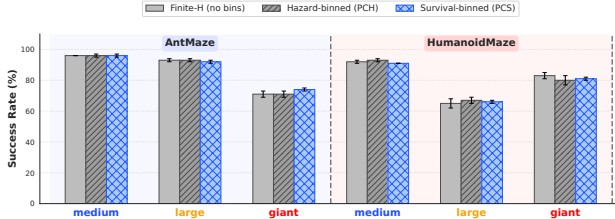

*Figure 2.* **Comparative analysis of the three estimators.** Success rates (%) on AntMaze and HumanoidMaze navigation tasks at increasing scales (medium/large/giant), comparing finite-horizon (no bins), hazard-binned (PCH), and survival-binned (PCS). The three variants perform comparably across all scales. Full numerical results are provided Tab. 4. The visualization scheme is adapted from (Ahn et al., 2025).

infinite-horizon variants. Both model the prefix on $[0, H]$ with $K$ bins and approximate the discounted tail $\sum_{t=H}^{\infty} \gamma^t S(t)$ using the final interval $[H, \infty)$.

**Piecewise-constant per-step hazard (PCH).** We assume the per-step hazard is constant within each bin:

$$h_\theta^\pi(t|s, g) = \bar{h}_\theta^\pi(k|s, g), \qquad t \in [b_k, b_{k+1}).$$

**Piecewise-constant survival per bin (PCS).** We assume the survival function is constant within each bin:

$$S_\theta^\pi(t|s, g) = S_\theta^\pi(b_k|s, g), \qquad t \in [b_k, b_{k+1}).$$

Per variant, we derive a grouped-time log-likelihood objective involving only the $K$ bin parameters and the corresponding plug-in estimator of the goal-conditioned value function. Their closed-form expressions are detailed in Appendix A.5.

**Hazard function architecture.** To efficiently parameterize temporal hazard, we propose a conditional low-rank basis selection that extends standard discrete-time survival frameworks (Fotso, 2018; Gensheimer & Narasimhan, 2019) and mixture-of-experts formulations (Lee et al., 2018; 2025). While traditional models often predict independent logits or rely on fixed parametric distributions (Nagpal et al., 2021) or static polynomial bases (Campanella et al., 2025), our architecture, illustrated in Figure 4, dynamically assembles a temporal structure by performing a double contraction between state-goal dependent weights and a learned basis library $\boldsymbol{\Psi} \in \mathbb{R}^{S \times H \times K}$. Specifically, given a state-goal encoding $z$, the model produces selection weights $\mathbf{w}(z) \in \mathbb{R}^S$ and mixing coefficients $\mathbf{c}(z) \in \mathbb{R}^K$, which are combined with the library to obtain hazard logits via the conditioned low-rank factorization $\boldsymbol{\ell}_{1:H}(z) = \mathbf{w}(z)^\top \boldsymbol{\Psi} \mathbf{c}(z)$. This approach allows the network to represent diverse temporal structures. We provide a more detailed explanation of the proposed architecture in Appendix A.4. Note that the hazard representation with a deep neural network remains an active line of research in the survival analysis community (Lee et al., 2018; 2025; Campanella et al., 2025).

## 4.4. Hierarchical policy for offline GCRL

So far, we have derived a survival-based objective for learning goal-conditioned value functions (SVL). While these estimators theoretically capture the task's underlying temporal structure, their practical utility is best demonstrated by their ability to guide an agent toward goals. To evaluate the quality of our representations in a decision-making context, we instantiate them within a learning algorithm. We propose **Hierarchical Survival Value Learning (HSVL)**, an offline GCRL algorithm that extracts a hierarchical policy from our SVL value estimator. We adopt a two-level architecture, following the HIQL approach (Park et al., 2023), in which a high-level policy selects subgoals to maximize the value, and a low-level policy executes actions to achieve subgoals.

**Policy extraction.** Given the learned survival value function $V_\theta$, we extract a hierarchical policy using advantage-weighted regression (AWR) (Peng et al., 2019), though the survival critic is also compatible with other offline RL policy-extraction schemes that exploit high-value actions (Tiofack et al., 2025). We learn a high-level subgoal policy $\pi_{\theta^h}(s_{t+k}|s_t, g)$ that proposes intermediate goals every $k$ steps, and a low-level action policy $\pi_{\theta^\ell}(a_t|s_t, s_{t+k})$ that executes actions to reach these subgoals. The parameters $\theta^h$ and $\theta^\ell$ are optimized by maximizing:

$$J_h(\theta^h) = \mathbb{E}_{(s_t, s_{t+k}, g)}\left[w_t^h \log \pi_{\theta^h}(s_{t+k}|s_t, g)\right], \qquad (16)$$

$$J_\ell(\theta^\ell) = \mathbb{E}_{(s_t, a_t, s_{t+1}, s_{t+k})}\left[w_t^\ell \log \pi_{\theta^\ell}(a_t|s_t, s_{t+k})\right] \quad (17)$$

The weights $w$ represent the advantage of the transition, computed using value differences:

$$w_t^h = \exp\left(\beta\left(V_\theta(s_{t+k}, g) - V_\theta(s_t, g)\right)\right), \qquad (18)$$

$$w_t^\ell = \exp\left(\beta V_\theta(s_{t+1}, s_{t+k}) - V_\theta(s_t, s_{t+k})\right), \qquad (19)$$

where $\beta > 0$ is an inverse-temperature parameter and with $V_\theta$ computed via the survival identity (9). The complete procedure is summarized in Algorithm 1.

**Design choices.** We choose this specific architecture for three key reasons. First, it isolates the contribution of our survival-based estimator. Because policy extraction depends only on the plug-in value $V_\theta$, any performance improvements can be directly attributed to the quality of the value estimation rather than additional algorithmic choices (e.g., target networks, or delicate actor-critic update schedules).

Second, a value-only interface is particularly robust for offline learning. Unlike $Q$-learning, which suffers from bootstrapping errors on out-of-distribution actions, our approach relies on value differences computed over in-distribution dataset transitions, providing a stable signal for advantage-weighted regression (Park et al., 2023).

Third, this design leverages hindsight relabeling as a standalone proxy for optimality. Prior work indicates that relabeling can drive goal-reaching behavior without explicit

**Algorithm 1** Hierarchical Survival Value Learning

1: **Input:** offline dataset $\mathcal{D}$, learning rates $\eta_h, \lambda_h, \lambda_\ell$
2: Initialize hazard $h_\theta$, high- and low-level policy $\pi_{\theta^h}, \pi_{\theta^\ell}$

    // Train the hazard function using Eq.(11)

3: **while** not converged **do**
4:     Sample $(s, g, \tau, c, \delta) \sim \mathcal{D}$
5:     $\theta \leftarrow \theta - \eta_h \nabla_\theta \mathcal{L}(\theta)$
6: **end while**

    // Train the high-level policy using Eq. (16)

7: **while** not converged **do**
8:     Sample $(s_t, s_{t+k}, g) \sim \mathcal{D}$
9:     $V_\theta(s_t, g) \leftarrow -\sum_{j \geq 0} \gamma^j S_\theta(j|s_t, g)$
10:     $V_\theta(s_{t+k}, g) \leftarrow -\sum_{j \geq 0} \gamma^j S_\theta(j|s_{t+k}, g)$
11:     $\theta^h \leftarrow \theta^h + \lambda_h \nabla_{\theta^h} J_h(\theta^h)$
12: **end while**

    // Train the low-level policy using Eq. (17)

13: **while** not converged **do**
14:     Sample $(s_t, a_t, s_{t+1}, s_{t+k}) \sim \mathcal{D}$
15:     $V_\theta(s_t, s_{t+k}) \leftarrow -\sum_{j \geq 0} \gamma^j S_\theta(j|s_t, s_{t+k})$
16:     $V_\theta(s_{t+1}, s_{t+k}) \leftarrow -\sum_{j \geq 0} \gamma^j S_\theta(j|s_{t+1}, s_{t+k})$
17:     $\theta^\ell \leftarrow \theta^\ell + \lambda_\ell \nabla_{\theta^\ell} J_\ell(\theta^\ell)$
18: **end while**
19: **Output:** $(\pi_{\theta^h}, \pi_{\theta^\ell})$

Bellman updates (Ghosh et al., 2021; Eysenbach et al., 2022). Accordingly, our method focuses on learning the survival value of the dataset trajectories, avoiding the risky action-maximization steps of Q-learning. Empirically, this formulation outperforms complex baselines, providing evidence that hindsight relabeling implicitly produces a value function superior to the behavior policy. This verifies that safe policy extraction is possible through hazard-based evaluation, without the extrapolation risks of typical offline RL.

# 5. Experiments

We evaluate our HSVL on the challenging OGBench benchmark (Park et al., 2025a), which includes high-dimensional state-based and visual offline goal-conditioned RL tasks. Our experiments aim to answer four key questions: (i) Can SVL scale to long-horizon, sparse-reward tasks where standard TD learning typically fails? (ii) Does SVL generalize to high-dimensional visual observations? (iii) How much of HSVL's performance is attributable to the survival critic itself, as opposed to the hierarchical policy extraction? (iv) How sensitive is the performance to architectural choices (e.g., network depth, hazard function parameterization)?

We compare HSVL against six offline GCRL baselines: goal-conditioned behavioral cloning (**GCBC**) (Lynch et al., 2020; Ghosh et al., 2021), goal-conditioned implicit V, Q-learning (**GCIVL**, **GCIQL**) (Kostrikov et al., 2022; Park et al., 2023), quasimetric RL (**QRL**) (Wang et al., 2023),

contrastive RL (**CRL**) (Eysenbach et al., 2022), and hierarchical implicit Q-learning (**HIQL**) (Park et al., 2023). HIQL is the closest algorithmic baseline to HSVL as it shares the same hierarchical value-to-policy extraction template (Algorithm 1). It differs only in how the value function is learned, via TD-style bootstrapping rather than survival value learning. The HIQL vs. HSVL comparison directly isolates the contribution of the survival formulation, holding the hierarchical actor fixed. CRL is the closest non-TD baseline that avoids bootstrapping. To isolate the survival critic's contribution relative to this strong MC baseline, we introduce a flat variant of SVL (no hierarchy) and re-evaluate CRL with the same flat DDPG+BC actor, so that the two methods differ only in their critics (Sec. 5.2). We report the normalized success rate averaged over 4 seeds. SVL implementation will be released upon paper acceptance, together with the derived HSVL adapted from HIQL.

## 5.1. Main results

Tab. 1 summarizes the performance across state-based and visual domains; full results are provided in Tabs. 6, 7, and 8 of the Appendix. HSVL matches or exceeds the performance of prior methods on standard tasks with substantial gains on complex, long-horizon problems. Note that we fix the inverse temperature parameter, $\beta$, to 3 across all experiments. Likewise, we kept OGBench default values for all hyperparameters not specific to SVL, as reported in Tab. 3.

**Scaling to long horizons.** We show substantial improvements on tasks requiring extended temporal reasoning. On antmaze-giant-navigate, HSVL outperforms the strongest baseline (HIQL) by a clear margin ($74\%$ vs. $65\%$). This gap widens for the complex humanoidmaze-giant navigation task: HSVL achieves $\mathbf{81}\%$ success rate, whereas HIQL reaches only $12\%$, and non-hierarchical methods fail ($0$–$3\%$). Since HSVL and HIQL share the same hierarchical extraction template and differ only in how the value is learned, we read the widening gap on longer horizons as controlled evidence that survival-based MLE mitigates the compounding bootstrap errors of TD over long horizons, consistent with recent findings that 1-step TD errors grow sharply with horizon (Park et al., 2025b).

**Visual generalization.** We further evaluate HSVL on pixel-based benchmarks to test its ability to learn from high-dimensional observations. On visual-antmaze-giant, our method achieves $\mathbf{61}\%$ success, surpassing the best hierarchical baseline (HIQL, $6\%$) and contrastive baseline (CRL, $47\%$). Furthermore, on the challenging visual-humanoidmaze-medium, HSVL is the only method to extract meaningful signal ($\mathbf{22}\%$ and $\mathbf{15}\%$ resp for navigate and stitch), while all baselines fail ($0$–$1\%$). This suggests that the hazard-based objective provides a robust learning signal even when combined with convolutional encoders,

*Table 1.* **Results on state-based and visual-based offline goal-conditioned RL.** HSVL outperforms six baselines across benchmark tasks, achieving substantial gains on complex, long-horizon tasks like humanoidmaze-giant-navigate-v0 (highlighted lines). Mean results ± std over 4 seeds and we refer to the Appendix for the full results (Tabs. 6, 7, and 8, and training details Tab. 3).

| Environment | GCBC | GCIVL | GCIQL | QRL | CRL | HIQL | HSVL (ours) |
|---|---|---|---|---|---|---|---|
| antmaze-medium-navigate-v0 | 29 ± 4 | 72 ± 8 | 71 ± 4 | 88 ± 3 | 95 ± 1 | **96 ± 1** | **96 ± 1** |
| antmaze-large-navigate-v0 | 24 ± 2 | 16 ± 5 | 34 ± 4 | 72 ± 6 | 83 ± 4 | 91 ± 2 | **92 ± 1** |
| antmaze-giant-navigate-v0 | 0 ± 0 | 0 ± 0 | 0 ± 0 | 14 ± 3 | 16 ± 3 | 65 ± 5 | **74 ± 1** |
| antmaze-teleport-navigate-v0 | 26 ± 3 | 39 ± 3 | 35 ± 5 | 35 ± 5 | **53 ± 2** | 36 ± 4 | 50 ± 2 |
| antmaze-medium-stitch-v0 | 45 ± 11 | 44 ± 6 | 29 ± 6 | 59 ± 7 | 53 ± 6 | **94 ± 1** | 83 ± 2 |
| antmaze-large-stitch-v0 | 3 ± 3 | 18 ± 2 | 7 ± 2 | 18 ± 2 | 11 ± 2 | **67 ± 5** | 31 ± 2 |
| antmaze-giant-stitch-v0 | 0 ± 0 | 0 ± 0 | 0 ± 0 | 0 ± 0 | 0 ± 0 | **2 ± 2** | 0 ± 0 |
| antmaze-teleport-stitch-v0 | 31 ± 6 | **39 ± 3** | 17 ± 2 | 24 ± 5 | 31 ± 4 | 36 ± 2 | 38 ± 1 |
| humanoidmaze-medium-navigate-v0 | 8 ± 2 | 24 ± 2 | 27 ± 2 | 21 ± 8 | 60 ± 4 | 89 ± 2 | **91 ± 0** |
| humanoidmaze-large-navigate-v0 | 1 ± 0 | 2 ± 1 | 2 ± 1 | 5 ± 1 | 24 ± 4 | 49 ± 4 | **66 ± 1** |
| humanoidmaze-giant-navigate-v0 | 0 ± 0 | 0 ± 0 | 0 ± 0 | 1 ± 0 | 3 ± 2 | 12 ± 4 | **81 ± 1** |
| humanoidmaze-medium-stitch-v0 | 29 ± 5 | 12 ± 2 | 12 ± 3 | 18 ± 2 | 36 ± 2 | 88 ± 2 | **89 ± 1** |
| humanoidmaze-large-stitch-v0 | 6 ± 3 | 1 ± 1 | 0 ± 0 | 3 ± 1 | 4 ± 1 | 28 ± 3 | **34 ± 2** |
| cube-single-play-v0 | 6 ± 2 | 53 ± 4 | **68 ± 6** | 5 ± 1 | 19 ± 2 | 15 ± 3 | 14 ± 2 |
| puzzle-3x3-play-v0 | 2 ± 0 | 6 ± 1 | **95 ± 1** | 1 ± 0 | 3 ± 1 | 12 ± 2 | 54 ± 3 |
| visual-antmaze-medium-navigate-v0 | 11 ± 2 | 22 ± 2 | 11 ± 1 | 0 ± 0 | 94 ± 1 | 93 ± 4 | **95 ± 1** |
| visual-antmaze-large-navigate-v0 | 4 ± 0 | 5 ± 1 | 4 ± 1 | 0 ± 0 | 84 ± 1 | 53 ± 9 | **85 ± 2** |
| visual-antmaze-giant-navigate-v0 | 0 ± 0 | 1 ± 1 | 0 ± 0 | 0 ± 0 | 47 ± 2 | 6 ± 4 | **61 ± 1** |
| visual-antmaze-teleport-navigate-v0 | 5 ± 1 | 8 ± 1 | 6 ± 1 | 6 ± 3 | **48 ± 2** | 37 ± 2 | 41 ± 1 |
| visual-antmaze-teleport-stitch-v0 | 32 ± 3 | 1 ± 1 | 1 ± 0 | 1 ± 2 | 32 ± 6 | **37 ± 4** | 25 ± 5 |
| visual-humanoidmaze-medium-navigate-v0 | 0 ± 0 | 0 ± 0 | 0 ± 0 | 0 ± 0 | 1 ± 0 | 0 ± 0 | **22 ± 2** |
| visual-humanoidmaze-medium-stitch-v0 | 1 ± 0 | 0 ± 0 | 0 ± 0 | 0 ± 0 | 1 ± 0 | 0 ± 0 | **15 ± 1** |

without requiring the explicit representation-learning losses used by prior visual RL methods.

**Why does HSVL perform better on humanoidmaze-giant than on humanoidmaze-large?** Our HSVL achieves a higher success rate on humanoidmaze-giant-navigate (81%) than on the smaller variant, humanoidmaze-large-navigate (66%). These results look counterintuitive, as the giant maze requires up to 3000 steps to reach the goal. We conjecture that this gap is driven primarily by dataset size rather than by the challenge of long-horizon planning. Although both datasets contain 1000 episodes, the giant variant includes 4M transitions, whereas the large one includes 2M transitions, yielding substantially more survival tuples after relabeling and thus more supervision for hazard learning. This interpretation is consistent with the MLE property of Lemma 4.2: increasing the number of samples reduces value estimation error at the canonical $\mathcal{O}(n^{-1/2})$ statistical rate.

### 5.2. Isolating the survival critic

To address question (iii), we evaluate the contribution of the survival critic without hierarchical policy extraction. The flat variant of SVL introduced above pairs the survival critic with a standard non-hierarchical DDPG+BC actor. This requires learning a state-action value $Q^\pi(s, a, g)$ rather than only $V^\pi(s, g)$. The corresponding derivation follows the same survival identity (Prop. 4.1), see Appendix A.3.

*Table 2.* **Flat-critic comparison on `navigate-v0`.** Success rates (%), mean ± std over 4 seeds. CRL (OGBench) reports original benchmark numbers, CRL (re-eval) and SVL use the same DDPG+BC actor (depth 6), isolating the critic. HSVL adds hierarchical extraction on top of SVL. Full results in Tab. 5

| | CRL (OGBench) | CRL (re-eval) | SVL (ours) | HSVL (ours) |
|---|---|---|---|---|
| antmaze-medium | 95 | 96 ± 1 | 96 ± 1 | 96 |
| antmaze-large | 83 | 89 ± 1 | 91 ± 0 | 92 |
| antmaze-giant | 16 | 50 ± 2 | 43 ± 2 | 74 |
| humanoidmaze-medium | 60 | 62 ± 0 | 88 ± 3 | 91 |
| humanoidmaze-large | 24 | 42 ± 1 | 66 ± 2 | 66 |
| humanoidmaze-giant | 3 | 7 ± 0 | 45 ± 2 | 81 |

**Comparison protocol.** We aim at a minimal and controlled comparison of the *critics*. Therefore, we re-evaluate CRL with the same DDPG+BC actor as SVL (depth 6), so that the methods differ only in their critics rather than in actor implementations from different codebases. We use CRL as the most directly comparable baseline. Like SVL, it is a non-TD, Monte-Carlo-style method, so contrasting the two critics under a shared actor isolates what the distributional survival formulation adds over scalar contrastive regression.

**Results.** Table 2 reports success rates across six `navigate-v0` tasks. The flat SVL critic outperforms the re-evaluated CRL on the harder settings, most visibly on humanoidmaze-large (66% vs. 42%) and humanoidmaze-giant (45% vs. 7%), while matching it on the easier ones.

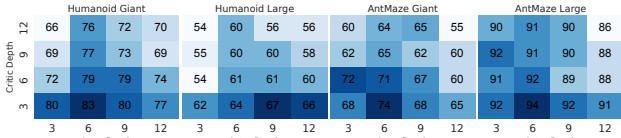

*Figure 3.* **Network depth ablation study.** Success rates (%) when varying actor depth and critic depth.

This indicates that the distributional, survival-based critic alone provides a stronger learning signal than contrastive regression in an identical flat setup. Adding hierarchical extraction (HSVL) compounds this gain on the giant mazes, while not necessary for medium and large sizes.

### 5.3. Architectural ablation studies

We address question (iv) by examining the sensitivity of HSVL to architectural design choices.

**Network depth.** Fig. 3 reports performance as we vary the MLP depth of both actor and hazard networks from 3 to 12 layers. On AntMaze, performance is remarkably stable across all configurations, indicating that HSVL is robust to network depth in this regime. On the higher-dimensional HumanoidMaze, we observe a benefit from increased capacity, with success rates peaking at depth 6. Overall, the results suggest that HSVL does not require excessive tuning regarding network architecture to achieve strong performance.

**Hazard function architecture.** We further investigate the impact of the discretization scheme used to model the time-to-goal. We compare the three estimators derived in Section 4.3: the finite-horizon model, the Piecewise-Constant Hazard (PCH), and the Piecewise-Constant Survival (PCS). As shown in Fig. 2, all three variants yield comparable success rates on OGBench (Park et al., 2025a) antmaze and humanoidmaze tasks. This robustness implies that the performance gains of HSVL stem from the survival formulation itself rather than a specific approximation method. Consequently, we adopt the PCS model for our main results due to its simplicity of implementation and efficiency.

## 6. Discussion and Conclusion

We introduced SVL, a probabilistic framework that frames GCRL as survival analysis. By modeling time-to-goal, we derived an identity that expresses the goal-conditioned value function as a discounted sum of survival probabilities. This enables direct value estimation via MLE for both success events and right-censored trajectories, thereby avoiding the instability and sample inefficiency often associated with TD bootstrapping. By combining SVL with hierarchical actors, we proposed HSVL for offline GCRL.

Our empirical evaluation highlights several findings. First, HSVL mostly outperforms the strongest hierarchical TD-based baseline on complex, long-horizon tasks, supporting our central hypothesis that by modeling the full time-to-goal distribution, SVL mitigates the compounding error inherent to Bellman backups. Second, the observed performance improvements with larger dataset sizes, even in the most challenging environments, suggest that the proposed framework scales favorably in offline settings. Finally, ablation studies indicate that HSVL is robust to architectural choices.

We view this work as a foundational step toward reframing sparse-reward GCRL. Our primary objective was to validate the core potential of the survival perspective; as such, we instantiated SVL within a standard offline architecture (HSVL) to isolate the benefits of the estimator itself. Specific design choices, such as the network architecture and hierarchy, were not optimized and remain open to refinement. A central advantage of the proposed framework is that it learns the entire distribution of time-to-goal, rather than only its expectation. In the present work, the policy compresses this rich distributional signal into a scalar value estimate, leaving significant information unexploited. Leveraging this distribution more directly, for exploration, risk-sensitive control or uncertainty-aware planning, is a promising direction for future work.

In this regard, the next step is to formally connect survival-based value estimation with distributional RL (Bellemare et al., 2017). Unlike traditional distributional RL methods, which rely on temporal-difference equations, SVL exploits the structure of the goal-conditioned setting to avoid bootstrapping altogether, positioning it as a structured distributional Monte Carlo alternative for sparse GCRL. While this paper focuses on offline policy evaluation, extending survival-based learning objectives to online settings is another important direction. In such regimes, learned hazard functions could inform exploration strategies by reasoning about uncertainty in time-to-goal predictions, enabling more principled exploration. Likewise, the survival identity is exact for returns determined by the goal-hitting time. Extending SVL to arbitrary dense rewards via a sparse/dense critic decomposition is a natural next step. A complementary direction, particularly relevant to safety-critical robotics, is to extend SVL to a competing-risks model that distinguishes goal achievement from informative failure terminations, rather than treating both as ordinary right censoring.

Overall, treating goal-reaching as a survival event provides a statistically viable alternative to conventional GCRL methods. By replacing Bellman bootstrapping with likelihood-based estimation, SVL offers a robust, scalable path toward long-horizon decision-making.

## Impact Statement

This paper presents work whose goal is to advance the field of reinforcement learning. There are many potential societal consequences of our work, none of which we feel must be specifically highlighted here.

## Acknowledgments

This work has received support from the French government, managed by the National Research Agency, under the France 2030 program with the references Organic Robotics Program (PEPR O2R) and "PR[AI]RIE-PSAI" (ANR-23-IACL-0008). This research was funded, in part, by l'Agence Nationale de la Recherche (ANR), projects NIMBLE project (ANR-22-CE33-0008) and PEPR O2R - AS2 (ANR-22-EXOD-0006). The European Union also supported this work through the ARTIFACT project (GA no. 101165695) and the AGIMUS project (GA no. 101070165). The Paris Île-de-France Région also supported this work in the frame of the DIM AI4IDF. The authors gratefully acknowledge the support and resources provided by the CLEPS infrastructure at Inria Paris. This work was performed using HPC resources from the GENCI-IDRIS Jean-Zay cluster (Grant 2024-AD010616763). Views and opinions expressed are those of the author(s) only and do not necessarily reflect those of the funding agencies.

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

# A. Appendix

## A.1. Link between survival and hazard functions

The recursive formula expressing $S$ using $h$, Eq. 7, is proved below for completeness. As $T^\pi(s, g)$ is a discrete random variable:

$$S^\pi(t \mid s, g) = \Pr(T^\pi(s, g) > t) \tag{20}$$
$$= \Pr(T^\pi(s, g) \geq t + 1) \tag{21}$$
$$= \Pr(T^\pi(s, g) = t + 1) \tag{22}$$
$$+ \Pr(T^\pi(s, g) > t + 1)$$
$$= h^\pi(t + 1 \mid s, g) \Pr(T^\pi(s, g) \geq t + 1)$$
$$+ \Pr(T^\pi(s, g) > t + 1) \tag{23}$$
$$= h^\pi(t + 1 \mid s, g) S^\pi(t \mid s, g)$$
$$+ S^\pi(t + 1 \mid s, g) \tag{24}$$

Equality 23 is the Bayesian rule applied to $\{T^\pi(s, g) = t + 1\} = \{T^\pi(s, g) = t + 1\} \cap \{T^\pi(s, g) \geq t + 1\}$.

## A.2. Proof of Proposition 4.1

Consider the GCRL setting where the agent receives a per-step penalty until success, $r(s_t, a_t, g) = -\mathbf{1}_{\{\|\phi(s_{t+1}) - g\| \geq \epsilon\}}$, and the episode terminates upon goal achievement.

$$V^\pi(s, g) = \mathbb{E}_{\tau \sim \rho^\pi(\cdot|g)} \left[ \sum_{t=0}^{\infty} \gamma^t r(s_t, a_t, g) \mid s_0 = s \right] \tag{25}$$

$$= -\mathbb{E}_{\tau \sim \rho^\pi(\cdot|g)} \left[ \sum_{t=0}^{\infty} \gamma^t \mathbf{1}_{\{\|\phi(s_{t+1}) - g\| \geq \epsilon\}} \mid s_0 = s \right] \tag{26}$$

$$= -\mathbb{E}_{T^\pi(s,g)} \left[ \sum_{t=0}^{T^\pi(s,g)-1} \gamma^t \right] \tag{27}$$

$$= -\mathbb{E}_{T^\pi(s,g)} \left[ \sum_{t=0}^{\infty} \gamma^t \mathbf{1}_{\{T^\pi(s,g) > t\}} \right] \tag{28}$$

$$= -\sum_{t=0}^{\infty} \gamma^t \mathbb{E}_{T^\pi(s,g)} \left[ \mathbf{1}_{\{T^\pi(s,g) > t\}} \right] \tag{29}$$

$$= -\sum_{t=0}^{\infty} \gamma^t \Pr(T^\pi(s, g) > t) \tag{30}$$

Equality 27 comes from the definition of $T^\pi(s, g)$.

## A.3. State-action survival identity (Q-function)

We extend the survival identity of Proposition 4.1 to the state-action value function. Define the first hitting time conditioned on an initial action $a_0 = a$ as

$$T^\pi(s, a, g) = \inf \left\{ t \geq 0 : \|\phi(s_{t+1}) - g\| < \epsilon \,\middle|\, \begin{matrix} s_0 = s,\, a_0 = a, \\ a_t \sim \pi(\cdot|s_t, g) \\ s_{t+1} \sim \mathcal{P}(\cdot|s_t, a_t) \end{matrix} \right\}. \tag{31}$$

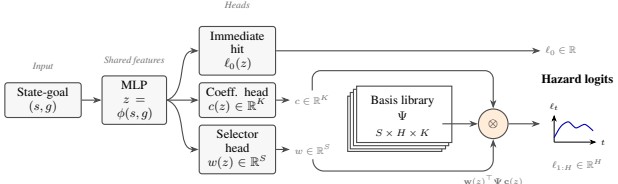

Figure 4. **Hazard function architecture.** From a given tuple of state $s$ and goal $g$, the network predicts time-to-goal behavior by combining a shared encoder with three heads: an immediate-hit predictor, coefficients describing temporal evolution, and weights that select among learned basis patterns. These components are combined to produce hazard predictions over time.

The associated survival function is

$$S^\pi(t|s, a, g) = \Pr(T^\pi(s, a, g) > t). \tag{32}$$

Under the same sparse-reward setting as Proposition 4.1, the goal-conditioned state-action value function satisfies

$$Q^\pi(s, a, g) = -\sum_{t=0}^{\infty} \gamma^t S^\pi(t \mid s, a, g). \tag{33}$$

*Proof.* The proof follows identically to that of Proposition 4.1 (Appendix A.2), replacing $T^\pi(s, g)$ with $T^\pi(s, a, g)$ throughout. The only difference is that the first action $a_0 = a$ is fixed rather than drawn from $\pi(\cdot|s_0, g)$, which does not affect the derivation.

## A.4. Explanation hazard architecture

There is a growing interest in the survival analysis community regarding the optimal deep neural architecture for modeling hazard functions. The emerging consensus suggests that independently predicting logits for each time bin is suboptimal, as it fails to exploit temporal correlations. Consequently, the primary design objective of modern approaches is to encode temporal information while maintaining computational efficiency (Lee et al., 2025; Campanella et al., 2025). For instance, the Dual Mixture-of-Experts framework (Lee et al., 2025) extracts features and combines them with learnable time embeddings through multiple heads. However, this approach may not scale, as maintaining separate heads for each time bin significantly increases the computational cost. Conversely, frameworks using static polynomial bases (e.g., Bernstein polynomials) (Campanella et al., 2025) offer a more computationally efficient alternative but impose a global, rigid inductive bias, assuming the underlying temporal structure is invariant across different inputs.

Our proposed architecture bridges these two paradigms by using Conditioned Low-Rank Factorization. To help understand our design, we first consider a simplified version with a single learnable basis set. For a given state-goal pair $(s, g)$,

we let $z = f_\theta(s, g)$ be the latent encoding. The hazard logit at time bin $t$ is expressed as:

$$\ell(t, s, g) = \mathbf{c}(z)^\top \boldsymbol{\psi}(t) + b_t, \tag{34}$$

where $\mathbf{c}(z) \in \mathbb{R}^K$ is a vector of predicted coefficients and $\boldsymbol{\psi}(t) \in \mathbb{R}^K$ is a learned temporal embedding (a basis function) for bin $t$. In that way, as in (Campanella et al., 2025), the hazard function (its logits) is a linear combination of a feature and a time embedding basis, but the time embedding is learnable as in the Dual Mixture-of-Experts.

However, a single basis set ($\boldsymbol{\psi}$) forces all $(s, g)$ pairs to share the same underlying temporal subspace, limiting the model's ability to represent fundamentally different survival behaviors (e.g., immediate success vs. long-term survival). To resolve this, we extend the formulation to include a Basis Library $\boldsymbol{\Psi} \in \mathbb{R}^{S \times H \times K}$ containing $S$ distinct basis sets. The model dynamically selects and mixes these sets using selection weights $w(z) \in \mathbb{R}^S$:

$$\ell(t, s, g) = \sum_{j=1}^{S} w_j(z) \left( \mathbf{c}(z)^\top \boldsymbol{\psi}_j(t) \right) + b_t. \tag{35}$$

By implementing this as a double tensor contraction-$\mathbf{w}(z)^\top \boldsymbol{\Psi} \mathbf{c}(z)$-we allow the model to dynamically "assemble" a custom temporal basis for every $(s, g)$ pair. This allows for better expressivity (capturing diverse "styles" of hazard functions) while maintaining the computational benefits of a single-head architecture, as the library $\boldsymbol{\Psi}$ is a shared, learned parameters across the entire dataset.

### A.5. Details on binned time modeling

**Piecewise-constant per-step hazard (PCH)** Let's assume the hazard function is constant within each bin:

$$h_\theta^\pi(t \mid s, g) = \bar{h}_\theta^\pi(k \mid s, g) \quad t \in [b_k, b_{k+1}).$$

For an observation $(s, g, \tau, c, \delta)$, we can write $\tau = b_{k_\tau} + m_\tau$ and $c = b_{k_c} + m_c$, where $k_\tau = \max\{k : b_k \leq \tau\}$, $k_c = \max\{k : b_k \leq c\}$, $m_\tau \in \{0, \ldots, L_{k_\tau} - 1\}$ and $m_c \in \{0, \ldots, L_{k_c}\}$. The grouped-time negative log-likelihood is

$$\mathcal{L}^{\text{PCH}}(\theta) = -\mathbb{E}_{(s,g,\tau,c,\delta)}[\delta \, \ell_\tau(\theta) + (1 - \delta) \, \ell_c(\theta)] \tag{36}$$

With

$$\ell_\tau^{\text{PCH}}(\theta) = \log \bar{h}_\theta^\pi(k_\tau \mid s, g) + \sum_{j < k_\tau} L_j \log(1 - \bar{h}_\theta^\pi(j \mid s, g))$$
$$+ m_\tau \log(1 - \bar{h}_\theta^\pi(k_\tau \mid s, g))$$

$$\ell_c^{\text{PCH}}(\theta) = \sum_{j < k_c} L_j \log(1 - \bar{h}_\theta^\pi(j \mid s, g))$$
$$+ m_c \log(1 - \bar{h}_\theta^\pi(k_c \mid s, g)).$$

Value evaluation admits a closed form for the prefix and a hazard-based tail. Let $q_{\hat\theta}(s, g) := \text{Pr}_{\hat\theta}(T^\pi(s, g) = 0)$ denote the predicted probability of immediate success, so that $S_{\hat\theta}^\pi(b_0 \mid s, g) = 1 - q_{\hat\theta}(s, g)$, and $S_{\hat\theta}^\pi(b_{k+1} \mid s, g) = S_{\hat\theta}^\pi(b_k \mid s, g)(1 - \bar{h}_{\hat\theta}^\pi(k \mid s, g))^{L_k}$. Then

$$V_{\hat\theta}^{\pi,\text{PCH}}(s, g) = -\sum_{k=0}^{K-1} \Delta_{\theta,k}(\gamma) S_{\hat\theta}^\pi(b_k \mid s, g)$$
$$- \frac{\gamma^H}{1 - \gamma(1 - h_K)} S_{\hat\theta}^\pi(H \mid s, g) \tag{37}$$

where $\Delta_{\theta,k}(\gamma) = \gamma^{b_k} \frac{1 - \left(\gamma(1 - \bar{h}_\theta^\pi(k|s,g))\right)^{L_k}}{1 - \gamma\left(1 - \bar{h}_\theta^\pi(k|s,g)\right)}$, $h_K = \bar{h}_{\hat\theta}^\pi(K \mid s, g)$ and the tail uses $\sum_{t=H}^\infty \gamma^t S(t) \approx \gamma^H S(H)/(1 - \gamma(1 - h_K))$.

**Piecewise-constant survival per bin (PCS).** We also consider the standard grouped-time model (Tutz et al., 2016) where the event time is observed at the bin level. Define the interval hazard

$$\tilde{h}_\theta^\pi(k \mid s, g) = \text{Pr}\left(T^\pi(s, g) \in [b_k, b_{k+1}) \,\middle|\, T^\pi(s, g) \geq b_k\right).$$

Survival at bin boundaries satisfies

$$S_\theta^\pi(b_{k+1} \mid s, g) = S_\theta^\pi(b_k \mid s, g)\left(1 - \tilde{h}_\theta^\pi(k \mid s, g)\right)$$

If an episode produces an event time $\tau \leq H$, we observe a binned event $[b_{k_\tau}, b_{k_\tau+1})$. If it is right-censored at $c \leq H$, we only observe survival up to the last completed boundary $b_{k(c)}$. The grouped-time negative log-likelihood is

$$\mathcal{L}^{\text{PCS}}(\theta) = -\mathbb{E}_{(s,g,\tau,c,\delta)}[\delta \, \ell_\tau(\theta) + (1 - \delta) \, \ell_c(\theta)] \tag{38}$$

With

$$\ell_\tau^{\text{PCS}}(\theta) = \log \tilde{h}_\theta^\pi(k(\tau) \mid s, g) + \sum_{j < k(\tau)} \log(1 - \tilde{h}_\theta^\pi(j \mid s, g))$$

$$\ell_c^{\text{PCS}}(\theta) = \sum_{j < k(c)} \log(1 - \tilde{h}_\theta^\pi(j \mid s, g))$$

As the survival function is constant within each bin, $S_\theta(t \mid s, g) \approx S_\theta(b_k \mid s, g)$ for $t \in [b_k, b_{k+1})$. Hence,

$$V_{\hat\theta}^{\pi,\text{PCS}}(s, g) = -\sum_{k=0}^{K-1} S_{\hat\theta}^\pi(b_k \mid s, g) \frac{\gamma^{b_k}\left(1 - \gamma^{L_k}\right)}{1 - \gamma}$$
$$- \frac{\gamma^H}{1 - \gamma} S_{\hat\theta}^\pi(H \mid s, g), \tag{39}$$

where the tail uses $\sum_{t=H}^\infty \gamma^t S(t) \approx \gamma^H S(H)/(1 - \gamma)$.

*Table 3.* **Hyperparameters**. Parameters not specific to the method are taken directly from OGBench (Park et al., 2025a).

| Hyperparameter | Value |
|---|---|
| Learning rate | 3e-4 |
| Optimizer | Adam |
| Minibatch size | 1024 (Maze), 256 (Visual env) |
| Total gradient steps | 1000000 (Maze), 500000 (Visual env) |
| MLP width | 512 |
| Critic depth | 3 |
| Actor depth | 6 |
| Activation function | GELU |
| Discount factor $\gamma$ | 0.995 (default), 0.999 (HumanoidMaze-giant), 0.99(maze-medium, manipulations) |
| Image augmentation probability | 0.5 (random crop) |
| Subgoal representation dimension | 256 |
| Inverse temperature $\beta$ | 3 |
| Policy $(p_{\text{cur}}^{\mathcal{D}}, p_{\text{traj}}^{\mathcal{D}}, p_{\text{rand}}^{\mathcal{D}})$ ratio | (0,1,0) (default), (0,0.5, 0.5) (stitch) |
| Number bin $K$ | 500 |
| Last interval edges $b_K$ | 10000 |
| subgoal step $k$ | 100 (humanoidmaze), 25(other) |
| Value $(p_{\text{cur}}^{\mathcal{D}}, p_{\text{traj}}^{\mathcal{D}}, p_{\text{rand}}^{\mathcal{D}})$ ratio | (0.08, 0.6, 0.32) |

*Table 4.* **Full results, hazard discretization comparison.** Success rates (%), mean $\pm$ std over 4 seeds, underlying Figure 2. We compare the three estimators from Section 4.3: finite-horizon (no bins), piecewise-constant per-step hazard (PCH), and piecewise-constant survival per bin (PCS).

| Environment | no bins | PCH | PCS |
|---|---|---|---|
| antmaze-medium-navigate-v0 | $96 \pm 0$ | $96 \pm 1$ | $96 \pm 1$ |
| antmaze-large-navigate-v0 | $93 \pm 1$ | $93 \pm 1$ | $92 \pm 1$ |
| antmaze-giant-navigate-v0 | $71 \pm 2$ | $71 \pm 2$ | $74 \pm 1$ |
| humanoidmaze-medium-navigate-v0 | $92 \pm 1$ | $93 \pm 1$ | $91 \pm 0$ |
| humanoidmaze-large-navigate-v0 | $65 \pm 3$ | $67 \pm 2$ | $66 \pm 1$ |
| humanoidmaze-giant-navigate-v0 | $83 \pm 2$ | $80 \pm 3$ | $81 \pm 1$ |

*Table 5.* **Full results for flat-critic comparison.** Per-task success rates (%), mean ± std over 4 seeds, on the six `navigate-v0` tasks from Table 2. CRL reports the original benchmark numbers (Park et al., 2025a) for reference. CRL (re-eval) and SVL share the same DDPG+BC actor (depth 6), so their comparison isolates the critic. Bold values indicate the best performance in each row.

| Environment | Task | CRL (Park et al., 2025a) | CRL (re-eval) | SVL (ours) |
|---|---|---|---|---|
| antmaze-medium-navigate-v0 | task1 | $97 \pm 1$ | $\mathbf{97 \pm 1}$ | $96 \pm 1$ |
| | task2 | $95 \pm 2$ | $\mathbf{97 \pm 1}$ | $95 \pm 0$ |
| | task3 | $92 \pm 3$ | $\mathbf{96 \pm 1}$ | $\mathbf{96 \pm 1}$ |
| | task4 | $94 \pm 5$ | $\mathbf{96 \pm 1}$ | $\mathbf{96 \pm 1}$ |
| | task5 | $96 \pm 2$ | $\mathbf{95 \pm 1}$ | $94 \pm 2$ |
| | overall | $95 \pm 1$ | $\mathbf{96 \pm 1}$ | $\mathbf{96 \pm 1}$ |
| antmaze-large-navigate-v0 | task1 | $91 \pm 3$ | $90 \pm 1$ | $\mathbf{93 \pm 1}$ |
| | task2 | $62 \pm 14$ | $83 \pm 2$ | $\mathbf{90 \pm 1}$ |
| | task3 | $91 \pm 2$ | $\mathbf{95 \pm 0}$ | $89 \pm 3$ |
| | task4 | $85 \pm 11$ | $90 \pm 2$ | $\mathbf{92 \pm 2}$ |
| | task5 | $85 \pm 3$ | $88 \pm 2$ | $\mathbf{94 \pm 1}$ |
| | overall | $83 \pm 4$ | $89 \pm 1$ | $\mathbf{91 \pm 0}$ |
| antmaze-giant-navigate-v0 | task1 | $2 \pm 2$ | $16 \pm 8$ | $\mathbf{18 \pm 4}$ |
| | task2 | $21 \pm 10$ | $\mathbf{62 \pm 5}$ | $46 \pm 5$ |
| | task3 | $5 \pm 5$ | $\mathbf{27 \pm 6}$ | $26 \pm 7$ |
| | task4 | $35 \pm 9$ | $\mathbf{65 \pm 4}$ | $57 \pm 4$ |
| | task5 | $16 \pm 10$ | $\mathbf{78 \pm 1}$ | $66 \pm 7$ |
| | overall | $16 \pm 3$ | $\mathbf{50 \pm 2}$ | $43 \pm 2$ |
| humanoidmaze-medium-navigate-v0 | task1 | $84 \pm 3$ | $\mathbf{88 \pm 1}$ | $\mathbf{88 \pm 4}$ |
| | task2 | $80 \pm 5$ | $93 \pm 2$ | $\mathbf{96 \pm 1}$ |
| | task3 | $43 \pm 11$ | $40 \pm 1$ | $\mathbf{92 \pm 5}$ |
| | task4 | $5 \pm 5$ | $0 \pm 1$ | $\mathbf{69 \pm 16}$ |
| | task5 | $87 \pm 7$ | $91 \pm 1$ | $\mathbf{97 \pm 0}$ |
| | overall | $60 \pm 4$ | $62 \pm 0$ | $\mathbf{88 \pm 3}$ |
| humanoidmaze-large-navigate-v0 | task1 | $36 \pm 11$ | $64 \pm 5$ | $\mathbf{74 \pm 3}$ |
| | task2 | $0 \pm 0$ | $0 \pm 0$ | $\mathbf{22 \pm 7}$ |
| | task3 | $54 \pm 17$ | $62 \pm 3$ | $\mathbf{83 \pm 6}$ |
| | task4 | $23 \pm 11$ | $54 \pm 1$ | $\mathbf{85 \pm 4}$ |
| | task5 | $6 \pm 4$ | $28 \pm 2$ | $\mathbf{64 \pm 7}$ |
| | overall | $24 \pm 4$ | $42 \pm 1$ | $\mathbf{66 \pm 2}$ |
| humanoidmaze-giant-navigate-v0 | task1 | $1 \pm 1$ | $1 \pm 0$ | $\mathbf{25 \pm 8}$ |
| | task2 | $9 \pm 5$ | $20 \pm 2$ | $\mathbf{46 \pm 4}$ |
| | task3 | $2 \pm 2$ | $1 \pm 0$ | $\mathbf{45 \pm 4}$ |
| | task4 | $3 \pm 2$ | $8 \pm 1$ | $\mathbf{45 \pm 1}$ |
| | task5 | $1 \pm 1$ | $3 \pm 1$ | $\mathbf{63 \pm 6}$ |
| | overall | $3 \pm 2$ | $7 \pm 0$ | $\mathbf{45 \pm 2}$ |

*Table 6.* **Full results, part 1**. The success rate is averaged over 4 random seeds. Bold values indicate the best performance in each row. Baseline performances are the official results provided by OGBench (Park et al., 2025a).

| Environment | Task | GCBC | GCIVL | GCIQL | QRL | CRL | HIQL | **HSVL** |
|---|---|---|---|---|---|---|---|---|
| antmaze-medium-navigate-v0 | task1 | $35 \pm 9$ | $81 \pm 10$ | $63 \pm 9$ | $93 \pm 2$ | $\mathbf{97 \pm 1}$ | $94 \pm 2$ | $96 \pm 1$ |
| | task2 | $21 \pm 7$ | $85 \pm 5$ | $78 \pm 8$ | $90 \pm 5$ | $95 \pm 2$ | $97 \pm 1$ | $\mathbf{98 \pm 1}$ |
| | task3 | $24 \pm 6$ | $60 \pm 13$ | $71 \pm 8$ | $86 \pm 6$ | $92 \pm 3$ | $96 \pm 2$ | $\mathbf{97 \pm 1}$ |
| | task4 | $28 \pm 7$ | $42 \pm 25$ | $59 \pm 12$ | $83 \pm 4$ | $94 \pm 5$ | $\mathbf{96 \pm 2}$ | $95 \pm 2$ |
| | task5 | $37 \pm 10$ | $92 \pm 3$ | $85 \pm 7$ | $88 \pm 8$ | $96 \pm 2$ | $96 \pm 2$ | $96 \pm 1$ |
| | overall | $29 \pm 4$ | $72 \pm 8$ | $71 \pm 4$ | $88 \pm 3$ | $95 \pm 1$ | $96 \pm 1$ | $96 \pm 1$ |
| antmaze-large-navigate-v0 | task1 | $6 \pm 3$ | $16 \pm 12$ | $21 \pm 6$ | $71 \pm 15$ | $91 \pm 3$ | $\mathbf{93 \pm 3}$ | $92 \pm 2$ |
| | task2 | $16 \pm 4$ | $5 \pm 6$ | $25 \pm 7$ | $77 \pm 7$ | $62 \pm 14$ | $78 \pm 9$ | $\mathbf{87 \pm 3}$ |
| | task3 | $65 \pm 4$ | $49 \pm 18$ | $80 \pm 5$ | $94 \pm 2$ | $91 \pm 2$ | $\mathbf{96 \pm 2}$ | $95 \pm 2$ |
| | task4 | $14 \pm 3$ | $2 \pm 2$ | $19 \pm 6$ | $64 \pm 8$ | $85 \pm 11$ | $\mathbf{94 \pm 2}$ | $93 \pm 2$ |
| | task5 | $18 \pm 4$ | $5 \pm 2$ | $26 \pm 9$ | $67 \pm 9$ | $85 \pm 3$ | $\mathbf{94 \pm 3}$ | $92 \pm 2$ |
| | overall | $24 \pm 2$ | $16 \pm 5$ | $34 \pm 4$ | $75 \pm 6$ | $83 \pm 4$ | $91 \pm 2$ | $\mathbf{92 \pm 1}$ |
| antmaze-giant-navigate-v0 | task1 | $0 \pm 0$ | $0 \pm 0$ | $0 \pm 0$ | $1 \pm 2$ | $2 \pm 2$ | $47 \pm 10$ | $\mathbf{66 \pm 3}$ |
| | task2 | $0 \pm 0$ | $0 \pm 0$ | $0 \pm 0$ | $17 \pm 5$ | $21 \pm 10$ | $74 \pm 5$ | $\mathbf{75 \pm 3}$ |
| | task3 | $0 \pm 0$ | $0 \pm 0$ | $0 \pm 0$ | $14 \pm 8$ | $5 \pm 5$ | $55 \pm 7$ | $\mathbf{66 \pm 3}$ |
| | task4 | $0 \pm 0$ | $0 \pm 0$ | $0 \pm 0$ | $18 \pm 6$ | $35 \pm 9$ | $69 \pm 5$ | $\mathbf{78 \pm 7}$ |
| | task5 | $1 \pm 1$ | $1 \pm 1$ | $1 \pm 1$ | $18 \pm 5$ | $16 \pm 10$ | $82 \pm 4$ | $\mathbf{84 \pm 3}$ |
| | overall | $0 \pm 0$ | $0 \pm 0$ | $0 \pm 0$ | $14 \pm 3$ | $16 \pm 3$ | $65 \pm 5$ | $\mathbf{74 \pm 1}$ |
| antmaze-teleport-navigate-v0 | task1 | $17 \pm 5$ | $35 \pm 5$ | $26 \pm 5$ | $31 \pm 6$ | $35 \pm 5$ | $\mathbf{37 \pm 5}$ | $\mathbf{37 \pm 4}$ |
| | task2 | $51 \pm 5$ | $41 \pm 5$ | $58 \pm 8$ | $47 \pm 22$ | $\mathbf{92 \pm 3}$ | $66 \pm 8$ | $\mathbf{92 \pm 2}$ |
| | task3 | $22 \pm 3$ | $36 \pm 8$ | $31 \pm 5$ | $35 \pm 6$ | $\mathbf{47 \pm 4}$ | $37 \pm 5$ | $45 \pm 3$ |
| | task4 | $25 \pm 5$ | $45 \pm 3$ | $33 \pm 5$ | $33 \pm 6$ | $\mathbf{50 \pm 2}$ | $30 \pm 2$ | $47 \pm 3$ |
| | task5 | $14 \pm 6$ | $38 \pm 6$ | $26 \pm 9$ | $28 \pm 8$ | $\mathbf{44 \pm 3}$ | $41 \pm 8$ | $27 \pm 3$ |
| | overall | $26 \pm 3$ | $39 \pm 3$ | $35 \pm 5$ | $35 \pm 5$ | $\mathbf{53 \pm 2}$ | $42 \pm 3$ | $50 \pm 2$ |
| antmaze-medium-stitch-v0 | task1 | $70 \pm 33$ | $76 \pm 13$ | $17 \pm 12$ | $43 \pm 20$ | $43 \pm 10$ | $\mathbf{92 \pm 2}$ | $85 \pm 5$ |
| | task2 | $65 \pm 19$ | $80 \pm 4$ | $22 \pm 16$ | $61 \pm 12$ | $46 \pm 14$ | $\mathbf{94 \pm 3}$ | $89 \pm 2$ |
| | task3 | $21 \pm 15$ | $16 \pm 12$ | $41 \pm 9$ | $72 \pm 29$ | $46 \pm 17$ | $\mathbf{95 \pm 2}$ | $\mathbf{95 \pm 2}$ |
| | task4 | $1 \pm 2$ | $0 \pm 0$ | $32 \pm 9$ | $80 \pm 9$ | $53 \pm 19$ | $\mathbf{93 \pm 2}$ | $54 \pm 11$ |
| | task5 | $70 \pm 33$ | $47 \pm 20$ | $34 \pm 14$ | $41 \pm 18$ | $75 \pm 8$ | $\mathbf{95 \pm 3}$ | $93 \pm 2$ |
| | overall | $45 \pm 11$ | $44 \pm 6$ | $29 \pm 6$ | $59 \pm 7$ | $53 \pm 6$ | $\mathbf{94 \pm 1}$ | $83 \pm 2$ |
| antmaze-large-stitch-v0 | task1 | $2 \pm 2$ | $23 \pm 9$ | $0 \pm 0$ | $7 \pm 5$ | $1 \pm 1$ | $\mathbf{85 \pm 5}$ | $13 \pm 12$ |
| | task2 | $0 \pm 0$ | $0 \pm 0$ | $0 \pm 0$ | $10 \pm 5$ | $4 \pm 4$ | $\mathbf{24 \pm 16}$ | $4 \pm 2$ |
| | task3 | $15 \pm 14$ | $69 \pm 6$ | $37 \pm 10$ | $73 \pm 8$ | $43 \pm 11$ | $\mathbf{94 \pm 3}$ | $69 \pm 15$ |
| | task4 | $0 \pm 0$ | $0 \pm 0$ | $0 \pm 0$ | $1 \pm 1$ | $5 \pm 5$ | $\mathbf{70 \pm 8}$ | $55 \pm 11$ |
| | task5 | $0 \pm 0$ | $0 \pm 0$ | $0 \pm 0$ | $1 \pm 1$ | $1 \pm 2$ | $\mathbf{60 \pm 9}$ | $13 \pm 7$ |
| | overall | $3 \pm 3$ | $18 \pm 2$ | $7 \pm 2$ | $18 \pm 2$ | $11 \pm 2$ | $\mathbf{67 \pm 5}$ | $31 \pm 2$ |
| antmaze-giant-stitch-v0 | task1 | $0 \pm 0$ | $0 \pm 0$ | $0 \pm 0$ | $0 \pm 0$ | $0 \pm 0$ | $0 \pm 1$ | $0 \pm 0$ |
| | task2 | $0 \pm 0$ | $0 \pm 0$ | $0 \pm 0$ | $0 \pm 0$ | $0 \pm 0$ | $\mathbf{5 \pm 5}$ | $0 \pm 0$ |
| | task3 | $0 \pm 0$ | $0 \pm 0$ | $0 \pm 0$ | $0 \pm 0$ | $0 \pm 0$ | $0 \pm 0$ | $0 \pm 0$ |
| | task4 | $0 \pm 0$ | $0 \pm 0$ | $0 \pm 0$ | $0 \pm 0$ | $0 \pm 0$ | $\mathbf{3 \pm 3}$ | $0 \pm 0$ |
| | task5 | $0 \pm 0$ | $0 \pm 0$ | $0 \pm 0$ | $2 \pm 2$ | $0 \pm 0$ | $0 \pm 1$ | $\mathbf{1 \pm 1}$ |
| | overall | $0 \pm 0$ | $0 \pm 0$ | $0 \pm 0$ | $0 \pm 0$ | $0 \pm 0$ | $\mathbf{2 \pm 2}$ | $0 \pm 0$ |
| antmaze-teleport-stitch-v0 | task1 | $21 \pm 13$ | $39 \pm 7$ | $12 \pm 4$ | $22 \pm 6$ | $30 \pm 6$ | $\mathbf{44 \pm 5}$ | $37 \pm 3$ |
| | task2 | $39 \pm 12$ | $\mathbf{44 \pm 6}$ | $18 \pm 7$ | $22 \pm 6$ | $30 \pm 4$ | $42 \pm 3$ | $\mathbf{44 \pm 2}$ |
| | task3 | $34 \pm 12$ | $\mathbf{36 \pm 8}$ | $18 \pm 4$ | $25 \pm 7$ | $23 \pm 11$ | $26 \pm 4$ | $\mathbf{36 \pm 3}$ |
| | task4 | $\mathbf{46 \pm 6}$ | $44 \pm 4$ | $18 \pm 5$ | $24 \pm 9$ | $38 \pm 4$ | $26 \pm 4$ | $35 \pm 2$ |
| | task5 | $16 \pm 14$ | $33 \pm 6$ | $17 \pm 6$ | $26 \pm 5$ | $32 \pm 7$ | $\mathbf{40 \pm 6}$ | $40 \pm 8$ |
| | overall | $31 \pm 6$ | $39 \pm 3$ | $17 \pm 2$ | $24 \pm 5$ | $31 \pm 4$ | $36 \pm 2$ | $\mathbf{38 \pm 1}$ |

*Table 7*. **Full results, part 2**. The success rate is averaged over 4 random seeds. Bold values indicate the best performance in each row. Baseline performances are the official results provided by OGBench (Park et al., 2025a).

| Environment | Task | GCBC | GCIVL | GCIQL | QRL | CRL | HIQL | **HSVL** |
|---|---|---|---|---|---|---|---|---|
| humanoidmaze-medium-navigate-v0 | task1 | $4 \pm 1$ | $22 \pm 5$ | $23 \pm 6$ | $12 \pm 7$ | $84 \pm 3$ | $\mathbf{95 \pm 2}$ | $92 \pm 1$ |
| | task2 | $8 \pm 4$ | $42 \pm 8$ | $49 \pm 6$ | $25 \pm 8$ | $80 \pm 5$ | $\mathbf{96 \pm 2}$ | $93 \pm 1$ |
| | task3 | $12 \pm 3$ | $15 \pm 3$ | $12 \pm 6$ | $25 \pm 10$ | $43 \pm 11$ | $79 \pm 6$ | $\mathbf{88 \pm 2}$ |
| | task4 | $2 \pm 1$ | $0 \pm 0$ | $1 \pm 0$ | $16 \pm 7$ | $5 \pm 5$ | $75 \pm 6$ | $\mathbf{89 \pm 2}$ |
| | task5 | $12 \pm 4$ | $40 \pm 8$ | $51 \pm 8$ | $29 \pm 12$ | $87 \pm 7$ | $\mathbf{97 \pm 1}$ | $94 \pm 2$ |
| | overall | $8 \pm 2$ | $24 \pm 2$ | $27 \pm 2$ | $21 \pm 8$ | $60 \pm 4$ | $89 \pm 2$ | $\mathbf{91 \pm 0}$ |
| humanoidmaze-large-navigate-v0 | task1 | $1 \pm 1$ | $6 \pm 2$ | $3 \pm 2$ | $3 \pm 2$ | $36 \pm 11$ | $67 \pm 4$ | $\mathbf{73 \pm 3}$ |
| | task2 | $0 \pm 0$ | $0 \pm 0$ | $0 \pm 0$ | $0 \pm 0$ | $0 \pm 0$ | $2 \pm 3$ | $\mathbf{8 \pm 6}$ |
| | task3 | $3 \pm 1$ | $6 \pm 2$ | $5 \pm 2$ | $17 \pm 6$ | $54 \pm 17$ | $88 \pm 3$ | $\mathbf{93 \pm 3}$ |
| | task4 | $2 \pm 1$ | $0 \pm 0$ | $1 \pm 1$ | $4 \pm 2$ | $23 \pm 11$ | $42 \pm 11$ | $\mathbf{89 \pm 2}$ |
| | task5 | $1 \pm 1$ | $1 \pm 1$ | $1 \pm 1$ | $2 \pm 1$ | $6 \pm 4$ | $47 \pm 10$ | $\mathbf{68 \pm 7}$ |
| | overall | $1 \pm 0$ | $2 \pm 1$ | $2 \pm 1$ | $5 \pm 1$ | $24 \pm 4$ | $49 \pm 4$ | $\mathbf{66 \pm 1}$ |
| humanoidmaze-giant-navigate-v0 | task1 | $0 \pm 0$ | $0 \pm 0$ | $0 \pm 0$ | $0 \pm 0$ | $1 \pm 1$ | $13 \pm 7$ | $\mathbf{74 \pm 3}$ |
| | task2 | $0 \pm 0$ | $1 \pm 1$ | $1 \pm 1$ | $2 \pm 1$ | $9 \pm 5$ | $35 \pm 11$ | $\mathbf{76 \pm 3}$ |
| | task3 | $0 \pm 0$ | $0 \pm 0$ | $0 \pm 0$ | $0 \pm 0$ | $2 \pm 2$ | $11 \pm 4$ | $\mathbf{79 \pm 1}$ |
| | task4 | $0 \pm 0$ | $0 \pm 0$ | $0 \pm 0$ | $0 \pm 0$ | $3 \pm 2$ | $2 \pm 2$ | $\mathbf{83 \pm 2}$ |
| | task5 | $1 \pm 1$ | $0 \pm 0$ | $1 \pm 1$ | $2 \pm 1$ | $1 \pm 1$ | $2 \pm 2$ | $\mathbf{93 \pm 2}$ |
| | overall | $0 \pm 0$ | $0 \pm 0$ | $0 \pm 0$ | $1 \pm 0$ | $3 \pm 2$ | $12 \pm 4$ | $\mathbf{81 \pm 1}$ |
| humanoidmaze-medium-stitch-v0 | task1 | $20 \pm 7$ | $13 \pm 3$ | $12 \pm 3$ | $6 \pm 5$ | $27 \pm 7$ | $84 \pm 5$ | $\mathbf{90 \pm 2}$ |
| | task2 | $49 \pm 12$ | $7 \pm 2$ | $8 \pm 5$ | $13 \pm 4$ | $37 \pm 7$ | $\mathbf{94 \pm 2}$ | $94 \pm 1$ |
| | task3 | $24 \pm 8$ | $25 \pm 3$ | $20 \pm 7$ | $30 \pm 6$ | $40 \pm 4$ | $86 \pm 4$ | $\mathbf{91 \pm 2}$ |
| | task4 | $3 \pm 2$ | $1 \pm 1$ | $2 \pm 2$ | $18 \pm 5$ | $28 \pm 7$ | $\mathbf{86 \pm 4}$ | $79 \pm 5$ |
| | task5 | $49 \pm 8$ | $16 \pm 3$ | $18 \pm 7$ | $22 \pm 2$ | $49 \pm 5$ | $90 \pm 4$ | $\mathbf{94 \pm 1}$ |
| | overall | $29 \pm 5$ | $12 \pm 2$ | $12 \pm 3$ | $18 \pm 2$ | $36 \pm 2$ | $88 \pm 2$ | $\mathbf{89 \pm 1}$ |
| humanoidmaze-large-stitch-v0 | task1 | $3 \pm 4$ | $2 \pm 1$ | $1 \pm 1$ | $0 \pm 0$ | $0 \pm 0$ | $\mathbf{21 \pm 5}$ | $16 \pm 4$ |
| | task2 | $0 \pm 0$ | $0 \pm 0$ | $0 \pm 0$ | $0 \pm 0$ | $0 \pm 0$ | $\mathbf{5 \pm 2}$ | $1 \pm 1$ |
| | task3 | $20 \pm 11$ | $3 \pm 2$ | $1 \pm 1$ | $16 \pm 7$ | $13 \pm 3$ | $\mathbf{84 \pm 4}$ | $83 \pm 4$ |
| | task4 | $2 \pm 1$ | $1 \pm 1$ | $0 \pm 1$ | $1 \pm 1$ | $4 \pm 1$ | $19 \pm 4$ | $\mathbf{51 \pm 5}$ |
| | task5 | $2 \pm 2$ | $1 \pm 1$ | $0 \pm 0$ | $0 \pm 0$ | $3 \pm 1$ | $12 \pm 2$ | $\mathbf{18 \pm 4}$ |
| | overall | $6 \pm 3$ | $1 \pm 1$ | $0 \pm 0$ | $3 \pm 1$ | $4 \pm 1$ | $28 \pm 3$ | $\mathbf{34 \pm 2}$ |
| cube-single-play-v0 | task1 | $7 \pm 3$ | $57 \pm 6$ | $\mathbf{71 \pm 9}$ | $6 \pm 2$ | $20 \pm 6$ | $15 \pm 5$ | $16 \pm 1$ |
| | task2 | $5 \pm 2$ | $51 \pm 6$ | $\mathbf{71 \pm 6}$ | $5 \pm 2$ | $20 \pm 4$ | $16 \pm 5$ | $11 \pm 2$ |
| | task3 | $7 \pm 3$ | $55 \pm 6$ | $\mathbf{70 \pm 6}$ | $4 \pm 1$ | $21 \pm 6$ | $16 \pm 3$ | $20 \pm 4$ |
| | task4 | $4 \pm 2$ | $50 \pm 4$ | $\mathbf{61 \pm 8}$ | $4 \pm 2$ | $16 \pm 3$ | $14 \pm 5$ | $10 \pm 2$ |
| | task5 | $4 \pm 2$ | $52 \pm 6$ | $\mathbf{67 \pm 7}$ | $4 \pm 3$ | $15 \pm 3$ | $13 \pm 4$ | $12 \pm 4$ |
| | overall | $6 \pm 2$ | $53 \pm 4$ | $\mathbf{68 \pm 6}$ | $5 \pm 1$ | $19 \pm 2$ | $15 \pm 3$ | $14 \pm 2$ |
| puzzle-3x3-play-v0 | task1 | $5 \pm 1$ | $17 \pm 4$ | $\mathbf{99 \pm 2}$ | $3 \pm 2$ | $11 \pm 3$ | $29 \pm 4$ | $67 \pm 4$ |
| | task2 | $2 \pm 1$ | $4 \pm 2$ | $\mathbf{96 \pm 3}$ | $0 \pm 0$ | $2 \pm 1$ | $11 \pm 3$ | $58 \pm 7$ |
| | task3 | $1 \pm 1$ | $3 \pm 1$ | $\mathbf{95 \pm 1}$ | $0 \pm 0$ | $1 \pm 1$ | $7 \pm 3$ | $42 \pm 3$ |
| | task4 | $1 \pm 1$ | $3 \pm 1$ | $\mathbf{91 \pm 3}$ | $0 \pm 0$ | $2 \pm 1$ | $5 \pm 1$ | $46 \pm 3$ |
| | task5 | $1 \pm 0$ | $2 \pm 1$ | $\mathbf{94 \pm 2}$ | $0 \pm 0$ | $2 \pm 1$ | $8 \pm 3$ | $54 \pm 5$ |
| | overall | $2 \pm 0$ | $6 \pm 1$ | $\mathbf{95 \pm 1}$ | $1 \pm 0$ | $3 \pm 1$ | $12 \pm 2$ | $54 \pm 3$ |

*Table 8.* **Full results, part 3**. The success rate is averaged over 4 random seeds. Bold values indicate the best performance in each row. Baseline performances are the official results provided by OGBench (Park et al., 2025a).

| Environment | Task | GCBC | GCIVL | GCIQL | QRL | CRL | HIQL | **HSVL** |
|---|---|---|---|---|---|---|---|---|
| visual-antmaze-medium-navigate-v0 | task1 | $17 \pm 6$ | $30 \pm 7$ | $16 \pm 3$ | $0 \pm 0$ | $\mathbf{92 \pm 2}$ | $90 \pm 4$ | $\mathbf{92 \pm 1}$ |
| | task2 | $8 \pm 2$ | $21 \pm 6$ | $7 \pm 2$ | $0 \pm 0$ | $94 \pm 2$ | $92 \pm 7$ | $\mathbf{96 \pm 1}$ |
| | task3 | $17 \pm 1$ | $24 \pm 5$ | $16 \pm 4$ | $0 \pm 0$ | $\mathbf{98 \pm 1}$ | $94 \pm 4$ | $96 \pm 1$ |
| | task4 | $12 \pm 2$ | $21 \pm 3$ | $9 \pm 2$ | $0 \pm 0$ | $94 \pm 2$ | $94 \pm 2$ | $\mathbf{95 \pm 1}$ |
| | task5 | $4 \pm 2$ | $16 \pm 5$ | $6 \pm 2$ | $0 \pm 0$ | $94 \pm 2$ | $94 \pm 5$ | $\mathbf{96 \pm 2}$ |
| | overall | $11 \pm$ | $22 \pm 2$ | $11 \pm 1$ | $0 \pm 0$ | $94 \pm 1$ | $93 \pm 4$ | $\mathbf{95 \pm 1}$ |
| visual-antmaze-large-navigate-v0 | task1 | $3 \pm 1$ | $7 \pm 2$ | $4 \pm 3$ | $0 \pm 0$ | $78 \pm 5$ | $60 \pm 10$ | $\mathbf{82 \pm 4}$ |
| | task2 | $4 \pm 3$ | $4 \pm 1$ | $2 \pm 1$ | $0 \pm 0$ | $80 \pm 3$ | $28 \pm 9$ | $\mathbf{82 \pm 5}$ |
| | task3 | $4 \pm 2$ | $6 \pm 2$ | $4 \pm 1$ | $1 \pm 1$ | $90 \pm 3$ | $85 \pm 10$ | $\mathbf{96 \pm 1}$ |
| | task4 | $4 \pm 2$ | $5 \pm 3$ | $6 \pm 1$ | $0 \pm 1$ | $\mathbf{88 \pm 3}$ | $46 \pm 7$ | $84 \pm 4$ |
| | task5 | $4 \pm 2$ | $5 \pm 1$ | $4 \pm 2$ | $0 \pm 0$ | $\mathbf{83 \pm 2}$ | $44 \pm 10$ | $81 \pm 3$ |
| | overall | $4 \pm 0$ | $5 \pm 1$ | $4 \pm 1$ | $0 \pm 0$ | $84 \pm 1$ | $53 \pm 9$ | $\mathbf{85 \pm 2}$ |
| visual-antmaze-giant-navigate-v0 | task1 | $0 \pm 0$ | $0 \pm 0$ | $0 \pm 0$ | $0 \pm 0$ | $17 \pm 2$ | $2 \pm 1$ | $\mathbf{45 \pm 4}$ |
| | task2 | $1 \pm 1$ | $2 \pm 1$ | $1 \pm 1$ | $0 \pm 0$ | $\mathbf{73 \pm 9}$ | $12 \pm 8$ | $68 \pm 4$ |
| | task3 | $0 \pm 0$ | $0 \pm 0$ | $0 \pm 0$ | $0 \pm 0$ | $22 \pm 6$ | $2 \pm 3$ | $\mathbf{43 \pm 8}$ |
| | task4 | $0 \pm 1$ | $0 \pm 1$ | $0 \pm 0$ | $0 \pm 0$ | $47 \pm 5$ | $4 \pm 2$ | $\mathbf{65 \pm 4}$ |
| | task5 | $1 \pm 1$ | $2 \pm 3$ | $1 \pm 1$ | $0 \pm 1$ | $77 \pm 5$ | $13 \pm 11$ | $\mathbf{83 \pm 4}$ |
| | overall | $0 \pm 0$ | $1 \pm 1$ | $0 \pm 0$ | $0 \pm 0$ | $47 \pm 2$ | $6 \pm 4$ | $\mathbf{61 \pm 1}$ |
| visual-antmaze-teleport-navigate-v0 | task1 | $2 \pm 2$ | $6 \pm 1$ | $2 \pm 1$ | $3 \pm 2$ | $32 \pm 3$ | $32 \pm 5$ | $\mathbf{34 \pm 4}$ |
| | task2 | $6 \pm 3$ | $9 \pm 3$ | $9 \pm 2$ | $6 \pm 4$ | $\mathbf{73 \pm 8}$ | $40 \pm 6$ | $71 \pm 4$ |
| | task3 | $9 \pm 1$ | $12 \pm 3$ | $9 \pm 2$ | $10 \pm 4$ | $\mathbf{47 \pm 3}$ | $33 \pm 1$ | $39 \pm 2$ |
| | task4 | $10 \pm 2$ | $10 \pm 2$ | $8 \pm 3$ | $6 \pm 4$ | $\mathbf{50 \pm 4}$ | $44 \pm 5$ | $37 \pm 5$ |
| | task5 | $1 \pm 1$ | $3 \pm 1$ | $3 \pm 1$ | $4 \pm 2$ | $\mathbf{36 \pm 5}$ | $33 \pm 7$ | $26 \pm 5$ |
| | overall | $5 \pm 1$ | $8 \pm 1$ | $6 \pm 1$ | $6 \pm 3$ | $\mathbf{48 \pm 2}$ | $37 \pm 2$ | $41 \pm 1$ |
| visual-antmaze-medium-stitch-v0 | task1 | $80 \pm 4$ | $0 \pm 1$ | $0 \pm 0$ | $0 \pm 0$ | $33 \pm 4$ | $75 \pm 8$ | $\mathbf{93 \pm 2}$ |
| | task2 | $\mathbf{90 \pm 4}$ | $1 \pm 2$ | $0 \pm 0$ | $0 \pm 0$ | $69 \pm 5$ | $85 \pm 7$ | $88 \pm 3$ |
| | task3 | $69 \pm 18$ | $15 \pm 6$ | $8 \pm 1$ | $0 \pm 0$ | $88 \pm 1$ | $\mathbf{92 \pm 1}$ | $82 \pm 7$ |
| | task4 | $1 \pm 1$ | $7 \pm 4$ | $3 \pm 1$ | $0 \pm 1$ | $70 \pm 12$ | $\mathbf{88 \pm 4}$ | $0 \pm 1$ |
| | task5 | $\mathbf{97 \pm 1}$ | $6 \pm 3$ | $1 \pm 1$ | $0 \pm 0$ | $85 \pm 5$ | $93 \pm 1$ | $83 \pm 15$ |
| | overall | $67 \pm 4$ | $6 \pm 2$ | $2 \pm 0$ | $0 \pm 0$ | $69 \pm 2$ | $\mathbf{87 \pm 2}$ | $69 \pm 3$ |
| visual-antmaze-teleport-stitch-v0 | task1 | $\mathbf{37 \pm 4}$ | $2 \pm 2$ | $1 \pm 1$ | $0 \pm 0$ | $20 \pm 5$ | $36 \pm 5$ | $36 \pm 15$ |
| | task2 | $36 \pm 3$ | $2 \pm 1$ | $1 \pm 1$ | $1 \pm 1$ | $40 \pm 9$ | $38 \pm 3$ | $\mathbf{42 \pm 3}$ |
| | task3 | $17 \pm 6$ | $2 \pm 1$ | $2 \pm 1$ | $3 \pm 4$ | $32 \pm 9$ | $\mathbf{36 \pm 5}$ | $8 \pm 10$ |
| | task4 | $39 \pm 9$ | $1 \pm 1$ | $0 \pm 0$ | $2 \pm 3$ | $\mathbf{45 \pm 7}$ | $37 \pm 6$ | $39 \pm 9$ |
| | task5 | $29 \pm 1$ | $1 \pm 1$ | $1 \pm 1$ | $1 \pm 1$ | $22 \pm 9$ | $\mathbf{38 \pm 5}$ | $3 \pm 2$ |
| | overall | $32 \pm 3$ | $1 \pm 1$ | $1 \pm 0$ | $1 \pm 2$ | $32 \pm 6$ | $\mathbf{37 \pm 4}$ | $25 \pm 5$ |
| visual-humanoidmaze-medium-navigate-v0 | task1 | $0 \pm 0$ | $0 \pm 0$ | $0 \pm 0$ | $0 \pm 0$ | $0 \pm 0$ | $0 \pm 0$ | $\mathbf{21 \pm 4}$ |
| | task2 | $0 \pm 0$ | $0 \pm 0$ | $0 \pm 0$ | $0 \pm 0$ | $0 \pm 0$ | $0 \pm 0$ | $\mathbf{14 \pm 7}$ |
| | task3 | $0 \pm 0$ | $0 \pm 0$ | $0 \pm 0$ | $0 \pm 0$ | $2 \pm 1$ | $0 \pm 1$ | $\mathbf{30 \pm 3}$ |
| | task4 | $0 \pm 0$ | $0 \pm 0$ | $0 \pm 0$ | $0 \pm 0$ | $0 \pm 0$ | $0 \pm 0$ | $\mathbf{13 \pm 2}$ |
| | task5 | $0 \pm 0$ | $0 \pm 0$ | $0 \pm 0$ | $0 \pm 0$ | $3 \pm 2$ | $0 \pm 0$ | $\mathbf{34 \pm 3}$ |
| | overall | $0 \pm 0$ | $0 \pm 0$ | $0 \pm 0$ | $0 \pm 0$ | $1 \pm 0$ | $0 \pm 0$ | $\mathbf{22 \pm 2}$ |
| visual-humanoidmaze-medium-stitch-v0 | task1 | $0 \pm 0$ | $0 \pm 0$ | $0 \pm 0$ | $0 \pm 0$ | $0 \pm 0$ | $0 \pm 0$ | $\mathbf{21 \pm 5}$ |
| | task2 | $0 \pm 0$ | $0 \pm 0$ | $0 \pm 0$ | $0 \pm 0$ | $0 \pm 0$ | $0 \pm 0$ | $\mathbf{18 \pm 3}$ |
| | task3 | $3 \pm 1$ | $0 \pm 0$ | $0 \pm 0$ | $0 \pm 0$ | $0 \pm 0$ | $0 \pm 0$ | $\mathbf{5 \pm 2}$ |
| | task4 | $0 \pm 0$ | $0 \pm 0$ | $0 \pm 0$ | $0 \pm 0$ | $\mathbf{3 \pm 2}$ | $1 \pm 2$ | $0 \pm 1$ |
| | task5 | $1 \pm 1$ | $0 \pm 0$ | $0 \pm 0$ | $0 \pm 0$ | $0 \pm 0$ | $0 \pm 0$ | $\mathbf{32 \pm 3}$ |
| | overall | $1 \pm 0$ | $0 \pm 0$ | $0 \pm 0$ | $0 \pm 0$ | $1 \pm 0$ | $0 \pm 0$ | $\mathbf{15 \pm 1}$ |

