# OpenReview forum: "SVL: Goal-Conditioned Reinforcement Learning as Survival Learning"
_ICML.cc/2026/Conference — ICML 2026 regular_

### Official Review · Reviewer_Wn4t · 2026-02-19

**Soundness:** 2
**Presentation:** 3
**Significance:** 2
**Originality:** 3
**Overall Recommendation:** 4
**Confidence:** 4

**Summary:**

This paper draws a connection between survival analysis and goal-conditioned RL to develop a new goal-conditioned value learning algorithm called SVL. Specifically, instead of directly fitting $V^\pi(s, g)$, the authors propose to fit a hazard function $h^\pi(t \mid s, g)$, which essentially measures the probability of reaching the goal at exactly time $t$ conditioned on the event of not reaching the goal until time $t-1$. They then convert the learned hazard function into a standard (behavioral) goal-conditioned value function using a mathematical relation between the two. Combined with a HIQL-like hierarchical policy extraction method, the authors show that the resulting algorithm outperforms standard offline goal-conditioned RL baselines on OGBench.

**Compliance With Llm Reviewing Policy:**

Affirmed.

**Final Justification:**

While it is still not entirely clear what the benefits of SVL are compared to standard distributional MC value learning (even after reading the second rebuttal), (1) I do think the new perspective introduced in this paper is valuable to the community and (2) my initial concerns about experimental results have mostly been addressed. Accordingly, I updated my rating from 3 to 4.

**Key Questions For Authors:**

I don't have additional questions other than the ones asked in the weakness section.

**Limitations:**

No, and I'd like to encourage the authors to include a discussion about limitations.

**Strengths And Weaknesses:**

[Strengths]

I think this is an interesting paper that provides a potentially new aspect to goal-conditioned RL. The main idea is somewhat conceptually similar to distributional RL -- instead of directly fitting a scalar value function, they fit some distributional information ("hazard function") about the probability of reaching the goal with a fixed policy $\pi$. To my knowledge, this perspective, and especially the connection to survival analysis, is novel. I can imagine that this new perspective may inspire future works in goal-conditioned RL. In terms of experiments, the authors provide adequate comparisons and ablations on standard OGBench tasks.

[Weaknesses]

While I generally like the new perspective introduced in the paper, I believe this paper has several weaknesses.
- While the connection to survival analysis is nice, it is a bit unclear to me why we *want* to use SVL instead of more traditional ways of fitting an MC value function. For example, the authors claim that SVL is potentially more scalable because it doesn't rely on TD learning, but why should we pick SVL over other MC-based methods (e.g., CRL, distance fitting, distributional MC value learning, etc.)?
- HSVL uses a hierarchical policy extraction method. However, most baselines considered in this work are not hierarchical (except HIQL). As the author mentioned in the paper, hierarchical policy extraction is largely orthogonal to the contribution of this paper, and I expected a fairer comparison between SVL (not HSVL) and the baselines. How does SVL perform on the OGBench tasks?
- It'd be a great plus to have an additional comparison with distribution MC value learning methods, given the conceptual similarity between SVL and distributional RL, as also acknowledged in Section 6.
- Compared to CRL (an MC goal-conditioned value learning algorithm), SVL is arguably more complex and has more hyperparameters (e.g., binning, the use of a special architecture, etc.) that potentially require tuning.
- Typos
    - L183 left: GRCL -> GCRL

I'd be happy to increase my score if these concerns are addressed.

---

> ### Author Rebuttal · Authors · 2026-03-31
>
> We warmly thank the reviewer for the thoughtful feedback. We appreciate the explicit note that the score could increase. We (1) clarify the positioning of SVL, (2) report a new non-hierarchical ablation comparing SVL and CRL under the same actor, and (3) discuss the complexity and limitations. **We compressed our answer due to rebuttal character limits; happy to expand.**
>
> **W1&3**
>
> We will revise the presentation to make SVL clearer as a **structured distributional Monte Carlo method for sparse GCRL**. In summary:
> - We are not aware of a prior distributional MC method for offline GCRL. We identified prior work on Distributional Monte Carlo Tree Search [1,2], but these approaches rely on online tree search and environment interaction.
> - Compared to non-distributional, non-TD methods (MC regression, distance fitting, CRL), the distinction is representational richness. While CRL learns a high-dimensional embedding space, the resulting critic is the scalar inner product in that space, and traditional MC methods regress a scalar expectation. In contrast, SVL learns the goal-hitting-time distribution to extract dense information that scalar methods may lack, as supported by the success of distributional methods in other RL setups [3].
> - Standard distributional RL (quantile or expectile methods) are based on TD updates [3] for dense rewards. While evaluating TD-based distributional methods on offline GCRL benchmarks would be interesting future work, theoretically, they inherit TD bootstrapping instability over long horizons.
>
> **W2**
>
> The reviewer is right that evaluating SVL without hierarchical extraction strengthens the comparison to non-hierarchical baselines. We therefore ran new experiments with a standard DDPG+BC actor and also re-evaluated CRL with the **same** actor (adapted from OGBench, with actor depth 6), isolating the critic as cleanly as possible.
>
> This ablation strongly supports the usefulness of the SVL critic even without hierarchy. On `humanoidmaze-giant-navigate`:
> - CRL (re-eval): 7
> - SVL (new): 45
> - HSVL (ours): 81
> - HIQL: 12
>
> Across all 6 `navigate-v0`:
> ||CRL(in OGBench)|CRL(re-eval)|SVL(new)|HSVL|
> |-|-|-|-|-|
> |antmaze-medium|95|96±1|96±1|96|
> |antmaze-large|83|89±1|91±0|92|
> |antmaze-giant|16|50±2|43±2|74|
> |humanoidmaze-medium|60|62±0|88±3|91|
> |humanoidmaze-large|24|42±1|66±2|66|
> |humanoidmaze-giant|3|7±0|45±2|81|
>
> Results are averaged over 4 seeds and hyperparameters below. To use a standard DDPG+BC actor, we adapted our formulation to compute $Q(s,a,g)$ rather than only $V(s,g)$. We will include the derivation and full per-task results in the Appendix.
>
> **W4**
>
> **Binning**
> We apologize if the submitted manuscript made SVL appear more complex than intended. By proposing several binning architectures, we aimed to explore practical estimators. Fig 2 was intended to show that, in practice, SVL is not brittle with respect to the estimation scheme. In the new experiment, we used the default estimator, PCS.
>
> **Network architecture**
> During the research phase, we explored different architectures but found they yielded similar results, so we left these details mostly in the Appendix. In the new non-hierarchical SVL experiment, to make the comparison with CRL fairest, we used a **standard MLP** mapping $(s,a)$ and $g$ to hazard logits, with the same number of layers and GeLU activation as in CRL.
>
> **Parameters**
> Goal sampling probabilities were fixed. For all 6 environments above $\gamma=0.995$. For the DDPG+BC actor, we used $\alpha=0.1$ for antmaze, following the CRL setup in OGBench, and $\alpha=0.03$ for humanoidmaze. We also tested CRL with $\alpha=0.03$ on humanoidmaze, which yielded approximately 64 / 38 / 5 on medium / large / giant, i.e. not better than $\alpha=0.1$.
>
> Our implementation runs at 300 it/s, compared to 400 it/s for CRL OGBench implementation. We will clean up small inefficiencies for the public release.
>
> **Limitations.**
> We agree that the current discussion section should be more explicit on the limitations. We already noted in the paper that the current policy compresses the learned distribution into a scalar value for action selection. We will add two limitations:
>
> 1) our method is best suited for terminate-on-success goal-conditioned RL problems. For arbitrary dense rewards, e.g. shaping terms or energy penalties, SVL should be combined with other methods, as further discussed in our response to Reviewer W4ml.
>
> 2) the architectural and computational overhead. While our ablations suggest that performance is reasonably robust to the additional design choices, the current implementation might not be the most efficient way to exploit the survival formulation. Developing lighter-weight parameterizations, adaptive discretizations, or continuous-time variants is an important direction for future work.
>
> ---
>
> [1] Hayes et al. (2023), Risk-Aware and Multi-Objective MCTS.
>
> [2] Dam et al. (2026), Distributional MCTS with Thompson Sampling.
>
> [3] Bellemare et al. (2017), Distributional RL.

---

> > ### Author Rebuttal · Reviewer_Wn4t · 2026-04-01
> >
> > Thanks for the response. I appreciate the new results, which would strengthen the paper. However, I'm not convinced by the argument "We are not aware of a prior distributional MC method for offline GCRL." After all, offline GCRL is a special case of RL, so we can just apply standard Monte-Carlo distributional value learning to this problem. I don't think we need a method name for this simple baseline. Moreover, on a separate note, there do exist distributional MC-based GCRL methods, such as "Search on the Replay Buffer: Bridging Planning and Reinforcement Learning", "Distance Weighted Supervised Learning for Offline Interaction Data", "Distributional Distance Classifiers for Goal-Conditioned Reinforcement Learning", and their related/cited works. Compared to distributional MC value learning, could the authors clarify what the main benefits of SVL are?

---

> > > ### Author Response · Authors · 2026-04-05
> > >
> > > We thank the reviewer for the continued engagement. We are glad the new non-hierarchical ablation was appreciated. Together with the clarifications on complexity and limitations, we believe W2 and W4 have been addressed. We now focus on W1 and W3.
> > >
> > > We thank the reviewer for sharing relevant prior work [1,2,3] that we will add to our paper. The reviewer is right that distributional MC methods can in principle be applied to offline GCRL. We identify DWSL [2] as a good representative of this approach and compare SVL to it below.
> > > The main benefits of our **structured** survival analysis formulation for GCRL are:
> > >
> > > 1. **Right-censoring.** Survival analysis provides a principled likelihood for cases where $g$ is not reached within the rollout (Eq 10) including cases where $g$ is sampled randomly from other trajectories
> > >
> > > 2. **Exact value recovery.** Prop 4.1 gives a closed-form identity for $V^\pi(s,g)$ compared to e.g. DWSL soft minimum-distance surrogate
> > >
> > > 3. **MLE guarantees.** The censored likelihood inherits consistency and asymptotic normality (Lemma 4.2)
> > >
> > >
> > > ## SVL vs DWSL
> > >
> > > DWSL and SVL both model the number of steps between states making DWSL the closest prior work. The key differences are:
> > >
> > > **Modelisation**
> > > -  DWSL learn a classifier $p^r_\theta (k\mid s,g)$ with $H$ classes where $k$ is the number of steps between $s$ and $g$ observed on a same trajectory
> > > -  SVL predicts $P(T^\pi(s,g)=t|T^\pi(s,g)\geq t)$ to get $S(t|s,g)=P(T^\pi(s,g)>t)$ including censored cases at rollout horizon $c$ with $c\leq H$
> > >
> > > SVL choice of conditional probabilities is rooted in survival analysis (Cox,1972,p.1) as it allows to account for cases where $g$ is not reached from $s$ within the observed horizon $c$. The OGBench paper [5] found it critical to evaluate $V(s,g)$ with $g$ sampled randomly with some probability $p^D_{rand}(g)$.
> > >
> > > DWSL only considers $(s,g)$ pairs belonging to a same trajectory. DWSL's code shows that the authors tried to treat unreached goals as belonging to the last bin of distance $H$ (catch-all bin). But the option was deactivated.
> > >
> > > **Extracting Value Function**
> > > Through survival estimates, SVL directly recovers
> > > $$V^\pi(s,g)=-\sum_{t=0}^{\infty}\gamma^t S^\pi(t\mid s,g).$$By contrast, after learning $p^r(k\mid s,g)$, DWSL constructs a soft minimum-distance surrogate
> > > $$\hat d_\alpha(s,g)=-\alpha \log\mathbb E_{k\sim p^r(\cdot\mid s,g)}\big[e^{-k/\alpha}\big].$$ The control signal for the policy in DWSL is a distance-based surrogate not the true discounted goal-conditioned value itself. To compare this surrogate metric to the value function, page 5 [2], DWSL authors make additional assumptions under which they prove only a lower bound Corollary 3.2.
> > >
> > > As **preliminary comparison**, we report results from the respective papers. Since OGBench antmaze-large uses the same layout as D4RL antmaze-large, this provides a reasonable reference point. We also include humamaze-giant, which is much harder and longer-horizon.
> > >
> > > |Method|DWSL [2]|CRL in [6]|CRL in [5]|CRL (rebuttal above)|HIQL [5]|HSVL (ours)|SVL (ours ablation)|
> > > |-|-|-|-|-|-|-|-|
> > > |D4RL antmaze-large-play|15|49|74||87|||
> > > |D4RL antmaze-large-diverse|18|54|79||87|||
> > > |OGBench antmaze-large navigate|||83|89|91|92|91|
> > > |OGBench humanoidmaze-giant|||3|7|12|81|45|
> > >
> > > While each paper uses its own implementation, the gaps are substantial and consistent.
> > >
> > > **D-NCE [3]**
> > >
> > > [3] extends CRL toward a distributional notion of goal occupancy, namely an $H$-dimensional vector whose entry at time $t$ is the probability of being at the goal at time $t$. D-NCE models goal occupancy, not first arrival. [3] faces the same issue as DWSL: handling goals that are not reached within the observed horizon. To address this, they use a catch-all bin. In SVL, this case is handled by right-censoring.
> > >
> > > We note technical concerns as pointed out in the [ICLR 2024 discussion](https://openreview.net/forum?id=qofh48zW3T), Proposition 1 does not appear to be correct. After carefully reading, we doubt Sec 4.2 & Appendix A correctness.
> > >
> > > **SoRB**
> > >
> > > We agree that [1] is relevant, as it uses distributional RL for GCRL. However, SoRB uses distributional TD Bellman update following [4] over short to mid range horizon. The core contribution of [1] is to combine RL and planning in a graph search, using the value function learned by distributional TD to estimate edge weights of nearby nodes in the graph. Then Dijkstra graph search is run on replay-buffer states to find long paths.
> > >
> > > We view graph-planning methods as complementary to our work. We will add this line of work in the related work section, including recent works (e.g. Graph-Assisted Stitching, Baek et al)
> > >
> > > ---
> > > [1] Eysenbach et al.(2019), Search on the Replay Buffer
> > >
> > > [2] Hejna et al.(2023), Distance Weighted Supervised Learning
> > >
> > > [3] Akella et al.(2023), Distributional Distance Classifiers for Goal-Conditioned Reinforcement Learning
> > >
> > > [4] Bellemare et al.(2017), Distributional RL
> > >
> > > [5] Seohong et al.(2025), OGBench
> > >
> > > [6] Eysenbach et al.(2023), Contrastive RL

---

### Official Review · Reviewer_UupA · 2026-03-04

**Soundness:** 3
**Presentation:** 3
**Significance:** 3
**Originality:** 3
**Overall Recommendation:** 5
**Confidence:** 3

**Summary:**

This paper proposes a new method based on survival analysis to learn the goal-conditioned value function for goal-conditioned RL. Traditionally, the goal-conditioned value function is learned using TD learning, where the value of a state needs to be propagated from the goal state via bootstrapping. The paper exploits the structure of goal-conditioned tasks and uses survival analysis to obtain a novel decomposition of the goal-conditioned value function. Particularly, the value function can be expressed as the discounted sum of survival functions, and survival functions can further be expressed in terms of hazard functions. Importantly, hazard functions can be learned by maximum likelihood estimation on a dataset, effectively turning the value function learning problem from a TD learning problem to a supervised learning problem, sidestepping the disadvantages of bootstrapping. Empirically, the paper demonstrates the superiority of such an approach compared to conventional goal-conditioned RL methods, especially in long-horizon, complex tasks where bootstrapping suffers.

**Compliance With Llm Reviewing Policy:**

Affirmed.

**Final Justification:**

I provided an initial rating of 5 (Accept), noting that the paper is strong across all dimensions with only minor concerns, which the rebuttal has been successful at fully addressing. I therefore am happy to keep this rating.

**Key Questions For Authors:**

- What does the $\pm$ numbers represent in Table 1? Confidence interval? Standard error? How do you compute it?
- Your Appendix formatting is wrong. Perhaps you accidentally commented out some line that starts the Appendix?
- If I understand correctly, HIQL is the most relevant baseline sharing almost the same algorithm as the proposed method (Algorithm 1), but the value function is learned using TD instead of survival value learning. If this is the case, I think it would be better to explicitly point this out when you're introducing the baselines you're comparing to (near line 341). This would make readers better understand why you're comparing with HIQL, and better see why the improved performance supports your claim (the claim mentioned in the first point of weaknesses).

Minor issues:

Polish
- Eq. 14: Formatting issue? $\theta$ is currently directly below $\Pr$ instead of as a subscript
- Fix heading of Section 6.3: "Explanation hazard architecture"
- Line 323: "where $\beta > 0$ is an inverse-temperature parameter" should be placed after Eq. 19 instead of after Eq. 17.
- Line 420: "is a promising direction for future work" missing period

Clarifying questions:
- Line 691: Let $S^\pi_{\hat \theta}(b_0 | s, g) = 1 - q_{\hat \theta}(s, g)$. What is $q_{\hat \theta}(s, h)$? Was it introduced earlier?

**Limitations:**

I don't see a discussion of the limitations of the proposed method. It'd be nice to add that to the discussion & conclusion section. For example, I see that you use number of bins $K=500$. With the basis library trick that you use, how expensive is the training of the hazard function and the value function estimation? How does it compare to standard methods where the neural network directly outputs the value?

**Strengths And Weaknesses:**

Strengths:
1. The paper is well-written, and concepts are explained clearly.
2. The paper is technically sound. Math is clear and easy to follow, and looks correct (I checked most except for Section 6.4). Claims are in general supported by empirical evidence (Table 1 shows strong performance of the proposed method).
3. The paper presents a novel framework for goal-conditioned RL that could inspire future research.

Weaknesses (note that these are mere suggestions, and do not affect my evaluation of acceptance):
1. In the experiments section, the paper claims that "by explicitly modeling the time-to-goal distribution via survival analysis, HSVL avoids the compounding bootstrap errors that plague TD learning over hundreds of steps." While this claim is in general supported by the strong performance of the proposed method in Table 1 compared to the baselines, a more direct and convincing way to support this claim is to directly quantify and compare the errors in the value function learned using TD and learned using survival value learning. I'm not sure if there's an easy way to do it though. Perhaps you can do it in some simple domains where the ground-truth value is easy to compute.
2. In Section 5.2, you vary the network depth, and show that the survival learning method is robust to different depths. Since you're only varying the network architecture, I find the claim "indicating that HSVL is not brittle to hyperparameter choices" is a bit of an over-claim.

---

> ### Author Rebuttal · Authors · 2026-03-31
>
> We thank the reviewer for the positive and careful evaluation. We are glad that the reviewer found the paper technically sound, clearly written, and potentially impactful. We also appreciate the concrete suggestions that further improve the paper.
>
> **Regarding Weakness 1.**
>
> We agree that a more direct quantification of value-estimation error would be interesting. Our current evidence is indirect but, in our view, fairly targeted. The strongest comparison in our experiments is with HIQL, since HSVL uses the same hierarchical extraction template and differs only in how the goal-conditioned value is learned. This was one of our motivations for adopting the HIQL-style architecture in the first place. With the policy-extraction pipeline fixed, the main difference is **TD-based value learning versus survival-based value learning**. We will make this point more explicit when introducing the baselines (see also Q3). Moreover, as mentioned in a closing note below, we did new experiments directly comparing SVL to CRL, enabling a more direct assessment of how the SVL critic scales relative to another Monte Carlo baseline.
>
> To add more supporting material, we will include the recent study of Park et al. [1], which identifies horizon as a central obstacle for TD-based offline RL and shows in controlled experiments that 1-step TD learning can have much larger Q-errors than n-step TD learning, with errors growing as the horizon increases (Fig. 6 & 7 in [1]).
> We view our results as complementary to that diagnosis. Rather than relying on n-step TD learning, SVL avoids bootstrapping by learning the goal-hitting-time distribution through a supervised, right-censored likelihood. The strong HSVL-vs-HIQL gap on the long-horizon tasks is currently our clearest evidence that avoiding Bellman propagation is beneficial.
>
> **Regarding Weakness 2.**
>
> We agree with your comment. Section 5.2 only varies the network depth, so the broader claim about hyperparameter robustness is too strong. We will revise the wording to the more precise statement that HSVL appears robust to network depth choices, rather than to hyperparameter choices in general.
>
> **Regarding Question 1.**
>
> Thank you for pointing this out. We will clarify explicitly that the reported $\pm$ values denote the standard deviation over the 4 random seeds used in evaluation.
>
> **Regarding Question 3.**
>
> Thank you for the suggestion and yes, you understood it correctly. We will revise the baseline introduction near line 341 to explicitly state that HIQL is the closest algorithmic baseline, as it shares the same hierarchical value-to-policy extraction template and learns the value function through TD-style bootstrapping rather than survival-value learning.
>
> **Regarding Question 2, minor issues, and clarifications.**
>
> Thank you for pointing these out. We will fix the broken Appendix formatting, the Eq. 14 formatting issue, the Section 6.3 heading, the placement of the inverse-temperature sentence around Eqs. (17)–(19), the missing period in the conclusion, and the notation clarification around line 691.
>
>
> **On limitations.**
>
> We agree that the current discussion should say more about limitations. We will add a brief discussion of the computational cost incurred by predicting a hazard over bins. In our implementation, the main additional cost is in the final output head, which predicts $K$ temporal quantities rather than a single scalar, while the hidden representation size remains comparable to the baselines. SVL is therefore slightly more expensive than a scalar-value head, but remains in the same practical regime as the other OGBench implementations. It can be noted that CRL predicts high-dimensional state embeddings $\phi(s,a)$ and $\psi(g)$ of size 512, so its network head is similarly sized.
>
>
> Overall, we are grateful for the reviewer’s positive assessment and constructive suggestions. The main revisions prompted by this review are to:
> (i) clarify that HIQL is the most relevant matched TD baseline,
> (ii) soften the wording around the network-depth ablation,
> (iii) explicitly define the uncertainty values in Table 1, and
> (iv) add a short discussion of computational cost and limitations.
>
> **Note on a complementary experiment conducted for another reviewer**. To address a concern raised by Reviewer Wn4t, we ran a new ablation study evaluating SVL with a standard DDPG+BC actor, rather than HSVL's hierarchical policy extraction, to isolate the scaling behavior of the SVL critic relative to other non-hierarchical policies. In particular, on `humanoidmaze-giant-navigate`, SVL alone achieves 45% success, compared to 7 for CRL and 12 for HIQL (and 81 for HSVL, with the hierarchical policies). For the full details, we refer to our rebuttal response to reviewer Wn4t.
>
> -------------
>
> [1] Park, S., Frans, K., Mann, D., Eysenbach, B., Kumar, A., & Levine, S. (2025). Horizon Reduction Makes RL Scalable. ArXiv, abs/2506.04168.

---

> > ### Author Rebuttal · Reviewer_UupA · 2026-04-01
> >
> > Thanks for the thoughtful response. I'm happy to keep my positive rating.

---

### Official Review · Reviewer_PX3A · 2026-03-13

**Soundness:** 2
**Presentation:** 2
**Significance:** 3
**Originality:** 3
**Overall Recommendation:** 4
**Confidence:** 3

**Summary:**

This paper deals with GCRL with a survival analysis framework, regarding the goal reaching time as a stopping time.
The authors parameterized a survival function (in the form of a hazard function) and directly mapped it into the value function, so that the GCRL without bootstrap can be achieved.

Using the practical approach in RL, i.e., censoring a.k.a. timeout truncation and general off-policy (or offline) sampling strategy, they showed that this satisfies the assumption for parametric MLE convergence.

Further, for the sample efficiency, they introduced hindsight relabeling, multi-timestep value estimation (Eq.15) with geometric time binning.

Finally, HIQL is approached within the learned GCRL framework.

**Compliance With Llm Reviewing Policy:**

Affirmed.

**Final Justification:**

Although there are some presentation issues, and a more detailed explanation among previous literature is required, I think only a minor revision is enough for this paper.

I had concern of the discounting sense, policy-dependent and policy-independent sense of the discounting. The authors explained this difference very clearly.

Therefore, I had changed my score from the rejection side to the acceptance side.

**Key Questions For Authors:**

**Q.1.** In Table 5., 'visual-antmaze-teleport-stitch-v0', other algorithms relatively uniform performance over tasks, while the proposed method HSVL shows polarized results across tasks. What is the property of this environment that induces such results?

**Q.2** What are the exact success rates underlying Figure 2?

**Limitations:**

Yes

**Strengths And Weaknesses:**

### **Strength**
This paper is well-written and easy to follow. Further, the direct philosophy of translating goal-reaching sparse reward to the dense reward without bootstrapping can be applied to many applications other than GCRL (e.g., safe RL with hazard state, reversing the goal-condition to the hazard-avoid).


### **Weakness**

**Weakness 1** Some novelty concerns:

The learned hazard function $h$ converges to the occupation measure of goal state as stated in [1, 2] and [3].

First, in [1], they see a hazard function as a generalized discount factor;
$$ V^{\pi}(s_{t0}; S) = \sum_{t=t0}^{\infty} S(t) r_{t}, $$ where  $s_{t0}$ is the state at time $t0$ and $S$ is the survival function at timestep $t$.
This exactly corresponds to the author's approach when $r_{t} = 1_{s =g}$. Meanwhile, [2] proposes an approach to learn a proper discount factor in continuous time MDP.

This view coincides with the fact that the discount factor is a stochastic interest in actuarial science [6].

Regarding [3], one may focus on the learned hazard function through Eq.(11).
It will exactly match with GAIL[4] discriminator-like objective, therefore, the hazard function will converge to the Jensen-Shannon potential of the goal state, by [3-5]. Ma et al. [3] exactly argued about this property; although they also showed that learning this potential is optional, the learning approach is included in their framework.

Considering two perspectives, please clearly state the difference between these two approaches and the proposed approach in the related work section.


**Weakness 2** Double discounting

As I mentioned in Weakness 1, the survival function itself has a discounting property (for the details, please read [1]). However, the authors also introduced a constant discount factor $\gamma \in (0, 1]$. This will distort the stopping time, resulting in different risk sensitivity (w.r.t goal reaching probability only, for the details see [6]).

Unlike standard RL, where discounting is, somehow, treated as a black-box hyperparameter, I think this view is important as the authors set the goal-reaching probability, thereby achieving the shortest path to the goal.

In consequence, I think they should state the impact of the discount factor in this paper and show the experiment result (maybe exclusively) or provide a theoretical justification. For the experiment result, a single fixed environment would be sufficient.

 **Weakness 3** Figure 2

In Figure 2, the ablation study does not seem to provide any statistically meaningful results among the binning strategies. I am not even sure whether it is a presentation issue or there is no difference in a statistical sense because the bar graph is squeezed.



**Minor**
In Table 1 and 5, bold notation is missing in 'visual-antmaze-teleport-stitch-v0'.

In Table 1, bold notation is missing in 'antmaze-teleport-stitch-v0'.



### **Refs.**
[1] Fedus, William, et al. "Hyperbolic discounting and learning over multiple horizons." arXiv preprint arXiv:1902.06865 (2019).

[2] Schultheis, Matthias, Constantin A. Rothkopf, and Heinz Koeppl. "Reinforcement learning with non-exponential discounting." Advances in neural information processing systems 35 (2022): 3649-3662.

[3] Ma, Jason Yecheng, et al. "Offline goal-conditioned reinforcement learning via $ f $-advantage regression." Advances in neural information processing systems 35 (2022): 310-323.

[4] Ho, Jonathan, and Stefano Ermon. "Generative adversarial imitation learning." Advances in neural information processing systems 29 (2016).

[5] Goodfellow, Ian J., et al. "Generative adversarial nets." Advances in neural information processing systems 27 (2014).

[6]  Wang, Shaun S. "A class of distortion operators for pricing financial and insurance risks." Journal of risk and insurance (2000): 15-36.

---

> ### Author Rebuttal · Authors · 2026-03-31
>
> We thank the reviewer for the thoughtful comments. We agree that the paper should better explain these connections and clarify the role of the discount factor. We also agree that [1], [2], and [3] are highly relevant related work, and we will discuss them more explicitly in the revision. At the same time, we would like to clarify that they are not equivalent to SVL.
>
> **W1.**
>
> **Hyperbolic discounting**.
> In [1] and [2], the key object is **policy-independent discounting scheme**, replacing the standard exponential discount $\gamma^t$ with another function. In [1], survival/hazard terminology is used to motivate alternative discount functions, especially the hyperbolic form $$
> s(t)=\Gamma_k(t)=\frac{1}{1+kt}.
> $$
> Therefore, the main challenge addressed in [1] is how to learn
> $$
> Q_{\pi}^{\Gamma_k}(s,a)=\mathbb{E}_{\pi}\left[\sum_t\Gamma_k(t)R(s_t,a_t)\mid s,a\right]
> \qquad \text{(Eq.12 in [1])}.
> $$
>
> Thus in [1], $s(t)$ directly replaces $\gamma^t$. Similarly, [2] studies continuous-time RL beyond exponential discounting. In both works, survival language is tied to **how future rewards are discounted**.
> By contrast, SVL does **not** modify the discounted GCRL objective. We use survival analysis to model the **policy-dependent first hitting time to the goal** $T^\pi(s,g)$ with survival function
> $$
> S^\pi(t\mid s,g)=\Pr(T^\pi(s,g)>t).
> $$
> This quantity characterizes the time-to-goal distribution under policy $\pi$ and is used for value estimation:
> $$
> V^\pi(s,g)=-\sum_{t=0}^{\infty}\gamma^t S^\pi(t\mid s,g).
> $$
> So in SVL, survival analysis is not a discounting rule; it is a statistical model of goal-reaching time under a fixed discounted objective. For this reason, we do not think it is accurate to say that SVL is equivalent to [1].
>
> In short, in [1] and [2], $s(t)$ is a **designer-chosen, discounting function**, whereas in SVL, $S^\pi(t\mid s,g)$ **characterizes a policy $\pi$**. These are different mathematical objects.
>
> **GoFAR.**
> In [3] the central object is discounted visitation distribution over states or state-action pairs, with objective $$
> \min_\pi D_{\mathrm{KL}}(d^\pi(s;g)\|p(s;g)),
> $$where $$ d^\pi(s,a;g)=(1-\gamma)\sum_{t=0}^{\infty}\gamma^t\Pr_\pi(s_t=s,a_t=a\mid g),\quad d^\pi(s;g)=\sum_a d^\pi(s,a;g)$$ By contrast, our Eq. (11) is a standard right-censored maximum-likelihood objective over **stopping times**, modeling the first-arrival distribution. Hence the optimized quantities are different: **discounted occupancy measures** (spatial) in GoFAR vs **first-passage-time distributions** (temporal) in SVL. For that reason, we do not think it is accurate to say that the SVL hazard converges to the occupation measure of the goal state, or that Eq. (11) is exactly a GAIL objective [4].
>
> **W2.**
>
> We agree our wording should be more precise. Our intention was **not** to introduce a second discount factor. In our formulation, $\gamma$ remains the standard discount parameter of the GCRL objective, while $S^\pi$ characterizes a given policy.
>
> The reviewer is right that $\gamma$ affects the induced objective. Maximizing value favors earlier goal achievement, but for fixed $\gamma<1$ this is more precisely a **discounted functional of the hitting-time distribution**, not the raw expected hitting time. We will revise the introduction accordingly and add a brief discussion of the induced risk sensitivity.
>
> Following the reviewer's suggestions, we ran a small ablation on the `antmaze-large-navigate` environment with 4 seeds:
>
> |$\gamma$|0.9|0.99|0.995|0.999|
> |-|-|-|-|-|
> ||31±4|92±1|93±1|90±1|
>
> This confirms that $\gamma$ matters in practice, as in most RL methods. Too small a discount hurts performance, while the values used in the paper perform well and consistently.
>
> **W3 / Q2**.
>
> We agree the explanation of Fig. 2 should be clearer. Our intention was not to claim a significant ordering among estimators, but rather their **robust comparability**. The main takeaway is that all three variants perform similarly, suggesting that HSVL’s gains come from the survival formulation itself rather than a particular approximation method. We will revise the caption to make this explicit. Due to the rebuttal character limit of 5000 characters, we omit the full per-task table here and will include the exact success rates in the appendix.
>
> **Question 1**
>
> Thank you for catching the missing bold formatting in Tables 1 and 5; we will fix it. Our current hypothesis is that the polarized results are driven mainly by the **stitching** property of the dataset rather than teleportation. We also observe higher task-to-task variability on another stitch environment, `visual-antmaze-medium-stitch` task 4, whereas this pattern does not appear on `visual-antmaze-teleport-navigate`. We will add a short discussion of this in Section 6.
>
>
> ---
> [1] Fedus et al. (2019), Hyperbolic Discounting.
>
> [2] Schultheis et al. (2022), Non-Exponential Discounting.
>
> [3] Ma et al. (2022), GoFAR.
>
> [4] Ho and Ermon (2016), GAIL.

---

> > ### Author Rebuttal · Reviewer_PX3A · 2026-04-01
> >
> > Dear Authors,
> > I sincerely appreciate the diligent and careful rebuttal. These answers fully resolved my concerns.
> >
> > I think only minor revisions will be required based on this discussion.
> >
> > Based on the authors' excellent explanations and their commitment to revising the presentation of the figures, I raise my score accordingly.

---

### Official Review · Reviewer_W4mL · 2026-03-15

**Soundness:** 3
**Presentation:** 4
**Significance:** 4
**Originality:** 4
**Overall Recommendation:** 5
**Confidence:** 4

**Summary:**

The paper introduces Survival Value Learning (SVL), reformulating Goal-Conditioned Reinforcement Learning (GCRL) as a survival analysis problem. By treating goal achievement as a survival event and episode truncation as right-censored data, the authors derive a closed-form identity showing that the goal-conditioned value function can be expressed precisely as a discounted sum of survival probabilities. This allows the model to learn a hazard function via Maximum Likelihood Estimation (MLE) over event and right-censored trajectories, circumventing the compounding instability of standard Bellman bootstrapping. SVL is combined with a hierarchical actor (HSVL) and evaluated on long-horizon OGBench tasks, demonstrating scaling.

**Compliance With Llm Reviewing Policy:**

Affirmed.

**Key Questions For Authors:**

1. Proposition 4.1 rigidly assumes a strict -1 per-step penalty. Is there a theoretical pathway to extend SVL to handle dense reward structures, or is the framework restricted to unshaped sparse-reward scenarios?
2. How does SVL adapt in environments where episode truncation is highly informative (e.g., catastrophic failure states), which explicitly violates the non-informative censoring assumption required for MLE?
3. Can you provide a formal upper bound on the value estimation error introduced by the quantization bias in the Piecewise-Constant Survival (PCS) geometric binning scheme, particularly for the infinite-horizon tail?

**Limitations:**

Yes. The authors acknowledge the computational complexity of the hazard architecture and note that the algorithm currently compresses rich distributional signals into scalar values.

**Strengths And Weaknesses:**

### Strengths
* Bypassing Bellman bootstrapping by mapping hitting times to hazard functions is elegant. Proposition 4.1 establishes an equivalence that directly addresses the compounding error problem in long-horizon offline RL.
* The formulation naturally incorporates incomplete trajectories (timeouts) as right-censored data, allowing the model to extract dense, valid MLE supervision from failed rollouts without relying on ad-hoc penalty shaping.
* The empirical results on OGBench are outstanding. Achieving 81% success on `humanoidmaze-giant` (compared to HIQL's 12%) validates the claim that SVL avoids error accumulation over long horizons.

### Weaknesses
* The closed-form equivalence (Proposition 4.1) requires a penalty of -1 at every step until the goal is achieved. This makes the method mathematically incompatible with environments that utilize dense shaping rewards, energy penalties, or varying action costs, limiting its real-world generalizability.
* MLE framework fundamentally relies on statistical assumption of non-informative right-censoring. In many RL scenarios, an episode terminating early is very informative (e.g., the agent fell over). The paper does not adequately address how informative censoring biases the hazard estimate.
* To scale to infinite horizons, authors apply Piecewise-Constant Survival (PCS) over geometrically increasing time bins. Forcing survival probabilities to remain constant across massive time bins at later timesteps will distort true multimodal density of arrival distribution.

---

> ### Author Rebuttal · Authors · 2026-03-31
>
> We thank the reviewer for the positive and technically thoughtful feedback. We are glad the reviewer appreciated both the survival-analysis reformulation and the strong long-horizon OGBench results. We also appreciate the questions about reward generality, censoring, and geometric binning, as they highlight assumptions and extensions that we should clarify.
>
> **W1/Q1.**
>
> We agree that Proposition 4.1 is exact only when the policy value is fully determined by the first goal-hitting time $T^\pi(s,g)$. This already covers a broader class than the constant $-1$ reward, for time-dependent penalties$$
> r_t(s_t,a_t,s_{t+1})=-\rho(t)\mathbf 1_{\{\|\phi(s_{t+1})-g\|>\epsilon\}}
> $$
> we obtain
> $$
> V^\pi(s,g)=-\sum_{t\ge 0}\gamma^t \rho(t)S^\pi(t\mid s,g)
> $$But for arbitrary dense rewards that depend on the full trajectory, such as shaping terms, energy costs, or action penalties, the return is not identifiable from the distribution of $T^\pi(s,g)$ alone; thus, SVL does not directly apply. A natural extension is to decompose the reward into a sparse goal-reaching term and a dense term,
> $$
> r_t=-\rho(t)\mathbf 1_{\|\phi(s_{t+1})-g\|>\epsilon}+r_{\mathrm{dense}}(s_t,a_t,s_{t+1})
> $$ and then learn two critics. An SVL critic for the time-to-goal component and a standard critic for the dense component, with the actor optimizing their combination. This fits naturally within multi-critic / multi-objective RL [1-2].
>
> **W2/Q2.**
>
> This is a very good point. We agree that if termination is informative, e.g. catastrophic failure, then treating it as ordinary right censoring violates the non-informative censoring assumption used in Lemma 4.2. Our intended setting is exogenous truncation due to a fixed rollout horizon/timeout, where the censoring time is imposed by the environment configuration rather than generated by the policy. For informative failure terminations, the appropriate extension is a competing-risks survival model [3] rather than ordinary censoring. One introduces an event type $E\in (\text{goal},\text{failure} )$ and terminal time
> $$
> T^\pi(s,g)=\min(T^\pi_{\text{goal}}(s,g),T^\pi_{\text{fail}}(s,g)),
> $$
> with cause-specific hazards
> $$
> h_k^\pi(t\mid s,g)=\Pr(T^\pi=t,E=k\mid T^\pi\ge t),\qquad k\in(\text{goal},\text{fail})
> $$
> and survival
> $$
> S^\pi(t\mid s,g)=\prod_{u=0}^{t}\bigl(1-h_{\text{goal}}^\pi(u\mid s,g)-h_{\text{fail}}^\pi(u\mid s,g)\bigr)
> $$
> We will clarify in the revision that our present consistency argument applies to exogenous horizon-based censoring, while informative failure termination requires an explicit multi-event extension. This is also a promising direction for safety-critical robotics.
>
>
> **W3/Q3.**
>
> We can indeed derive an upper bound on the value function error due to binning, between the true $V$ and the estimated $\widehat V$. First split at the last bin:
> $$|V^\pi(s,g)-\widehat{V}^\pi(s,g)|\le\sum_{t<b_K}\gamma^t|S^\pi(t\mid s,g)-\widehat{S}^\pi(t\mid s,g)|+\sum_{t=b_K}^{\infty}\gamma^t|S^\pi(t\mid s,g)-\widehat{S}^\pi(t\mid s,g)|.$$ Then partition the first sum following the bins
> $$
> 0=b_0<b_1<b_2<\cdots<b_K,
> \qquad
> I_k=\{b_k,\dots,b_{k+1}-1\},
> $$ The estimated $\widehat{S}^\pi(\cdot\mid s,g)$ is constant on each bin: $\forall t \in I_k$,
> $\widehat{S}^\pi(t\mid s,g)=c_k^\pi(s,g)$.
>
> And given that $t \mapsto S^\pi(t|s,g)$ is non-increasing, we have $$
> |S^\pi(t\mid s,g)-c_k^\pi(s,g)|
> \le
> \varepsilon_k^\pi(s,g),
> $$ where $$
> \varepsilon_k^\pi(s,g)
> :=
> \max\left(
> |S^\pi(b_k\mid s,g)-c_k^\pi(s,g)|,
> |S^\pi(b_{k+1}-1\mid s,g)-c_k^\pi(s,g)|
> \right).
> $$
>
> Finally, since $\widehat{S}^\pi(t\mid s,g)=\widehat{S}^\pi(b_K\mid s,g)$, for $t \geq b_K$: $$
> |V^\pi(s,g)-\widehat{V}^\pi(s,g)|
> \le
> \sum_{k = 0}^{K-1}\gamma^{b_k}
> \frac{1-\gamma^{b_{k+1}-b_k}}{1-\gamma}
> \varepsilon_k^\pi(s,g) + \frac{\gamma^{b_K}}{1-\gamma} \varepsilon_{last}^\pi(s,g),
> $$ where $$
> \varepsilon_{last}^\pi(s,g)
> :=
> \max\left(
> |S^\pi(b_K\mid s,g)-\widehat{S}^\pi(b_K\mid s,g)|,
> |(\lim_{t\to\infty} S^\pi(t\mid s,g))-\widehat{S}^\pi(b_K\mid s,g)|
> \right).
> $$ Overall, $\varepsilon_{last}^\pi(s,g) \leq 1$, and the contribution of the last bin is bounded by $\frac{\gamma^{b_K}}{1-\gamma}$. In practice, using $b_K=10000$ and $\gamma=0.995$, the contribution of the last bin is bounded by $\frac{\gamma^{b_K}}{1-\gamma} \simeq 3 \cdot 10^{-20}$.
> It may not be the tightest upper bound, but it supports the intuition that geometric binning is natural in the long-horizon regime. Although later bins become wider, their impact on the error is exponentially damped through the factor $\gamma^{b_k}$. Hence, approximation errors in the far tail are downweighted by discounting. Different modes within a single bin cannot be separated, but geometric binning preserves significantly different modes. We agree that making this point explicit would strengthen the paper.
>
> We compress due to rebuttal limits; happy to expand.
>
> ---
> [1] Mossalam et al. (2016), Multi-Objective Deep RL.
>
> [2] Mysore et al. (2022), Multi-Critic Actor Learning.
>
> [3] Lee et al. (2018), DeepHit.

---

### Decision · Program_Chairs · 2026-04-30

**Decision:**

Accept (regular)

**Comment:**

The reviewers agree that this is a well-executed and well-presented paper, which makes a solid contribution to an interesting reinforcement learning problem. The reviewers identify the following strengths:
1) The paper is clear, and the mathematical derivations are easy to follow.
2) The thorough experimental results show a significant improvement over baseline algorithms.
3) The approach is original.

Most of the serious concerns raised in the original reviews were addressed by the authors in their response. The reviewers agree that most of the remaining concerns are minor and do not preclude publishing the paper.